# A Bayesian updating framework for calibrating the hydrological parameters of road networks using taxi GPS data

Xiangfu Kong[1,2], Jiawen Yang[2], Ke Xu[3], Bo Dong[1], Shan Jiang[4,5]

[1] Research Center for AI Social Governance, Zhejiang Lab, Hangzhou, Zhejiang 311100, China
[2] Shenzhen Graduate School, Peking University, Shenzhen, Guangdong 518055, China
[3] Zhejiang Development and Planning Institute, Hangzhou, Zhejiang 310012, China
[4] Department of Urban and Environmental Policy and Planning, Tufts University, Medford, MA 02155, USA
[5] Department of Civil and Environmental Engineering, Tufts University, Medford, MA 02155, USA

*Correspondence to*: Xiangfu Kong (kongxf@zhejianglab.com)

**Abstract.** Hydrological parameters should pass through a careful calibration procedure before being used in a hydrological model that aids decision making. However, significant difficulty is encountered when applying existing calibration methods to regions in which runoff data are inadequate. To achieve accurate hydrological calibration for ungauged road networks, we propose a Bayesian updating framework that calibrates hydrological parameters based on taxi GPS data. Hydrological parameters were calibrated by adjusting their values such that the runoff generated by acceptable parameter sets corresponded to the road disruption periods during which no taxi points are observed. The proposed method was validated on 10 flood-prone roads in Shenzhen and the results revealed that the trends of runoff could be correctly predicted for 8 of 10 roads. This study demonstrates that the integration of hydrological models and taxi GPS data can provide viable alternative measures for model calibration to derive actionable insights for flood hazard mitigation.

## 1 Introduction

In the context of climate change and increased urbanization, flooding poses far-reaching threats to urban road networks of coastal metropolises (Balistrocchi et al., 2020). In Australia, approximately 53% of flood-related drowning deaths were the result of vehicles driving into flood waters. Additionally, indirect losses caused by flooding such as cancelled commutes, mandatory detours, and travel time delays often outweigh direct losses (Kasmalkar et al., 2020). Quantifying the impact of flood exposure requires the prediction of surface runoff over roads and road disruptions induced by runoff, which are critical for the implementation of  flood mitigation, traffic resilience improvement, and early warning systems.

Public concerns regarding road flooding hazards have created pressure to develop fine-grained and accurate models for hydrological simulation. Hydrological modeling is based on a relatively well-established theory that can provide approximations of real-world hydrological system and has been widely used in many road-related studies (Versini et al., 2010; Yin et al., 2016; Safaei-Moghadam et al., 2022). Because hydrological modeling is subject to uncertainty that arises

from the over-simplified reflection of hydrological systems, initial and boundary conditions, and lack of true knowledge, parameters for hydrological models must be carefully calibrated prior to their application to practical problems, so that models can closely match the historical trends (Gupta et al., 1998). As un-calibrated models are indefensible and sterile, very

few models documented in the literature have been applied without a calibration procedure (Beven, 2012).

Over the past four decades, numerous studies have been conducted on the development of calibration methods. Methodologies for model calibration range from simple trial-and-error methods that adjust one parameter value in each iteration until the differences between predicted and observed values are satisfactory to Bayesian updating framework that reject the concept of a single correct solution. To a great extent, the success of model calibration is dominated by the

availability of field-observed runoff data. However, runoff data are generally only gathered at a few sites and some cities never measure runoff data in built-up regions (Gebremedhin et al., 2020). Although runoff data can be effectively collected by administration departments in some cities, these cities are not always motivated to share these data with the public. For example, among China's top 10 largest cities[1], only Shenzhen has shared runoff-related data on an open data platform. For model calibration at the road scale, runoff data are even more difficult to acquire because road networks are far denser than

river networks and flood gauges are only installed in a few flood-prone roads based on their high measurement cost, leaving most roads ungauged. As pointed out by Beven (2012, p:55), "the ungauged catchment problem is one of the real challenges for hydrological modelers."

This lack of hydrological data has prompted researchers to seek additional data sources to support flood-related decision making. Based on the advancement of mobile telecommunication technologies, big data are emerging as alternative

sources of information for coping with flood risks (Paul et al., 2018; Li et al., 2018; Gebremedhin et al., 2020). Citizens can voluntarily or passively act as human sensors to generate georeferenced data to improve flood monitoring. Many studies have leveraged crowdsourced social media data (Brouwer and Eilander, 2017; Sadler et al., 2018; Zahura et al., 2020), mobile phone data (Yabe et al., 2018; Balistrocchi et al., 2020), and taxi GPS data (She et al., 2019; Kong et al., 2022). However, most previous works have concentrated on using big data either for flood mapping or mining spatiotemporal

patterns (Restrepo-Estrada et al., 2018), and parameter calibration for ungauged roads based on big data remains an open problem.

This study extends our previous study (Kong et al., 2022) by going a step further than simply recognizing flooded roads. We propose a calibration method for road-related hydrological parameters based on taxi GPS data. Many studies have shown that vehicle-related information during rainfall, including vehicle volume, speed, and trajectory information, is useful

for flooded road detection (Zhang et al., 2019; Qi et al., 2020; Yao et al., 2020). When a road segment is inundated by heavy rainfall, the vehicle volume may exhibit a sharp or gradual drop depending on the intensity of the rainfall event. Conversely, an abnormal drop in vehicle volume during the rainfall may imply that a road has experienced rainfall-induced inundation. This motivates us to use traffic-related data sources to calibrate hydrological parameters. In this study, we developed a

---

[1] Rank by the resident population in 2021.

transformation process that converts rainfall time series data into a time series of probabilities that no taxis will drive on a

road (no-taxi-passing probability hereafter) for a given hydrological parameter set. We then assigned a probability to every parameter set by integrating the no-taxi-passing probability with observed taxi GPS data. We outlined a generalized taxi-data-driven calibration framework and implemented a framework with specific hydrological and transportation models.

## 2 Methodology

### 2.1 Bayesian updating procedure

Observed data are not always as informative as expected and may be inconsistent with other data sources; hydrologists typically adopt the Bayesian framework to update hydrological parameters, which provides a generalized formalism that integrates prior probability representing prior knowledge with likelihood that reflects how accurately a model can reproduce observations to form a posterior probability. Suppose we have several versions of a hydrological model, each with different sets of parameters. Then, the purpose of the Bayesian updating procedure adopted in this study is to assign a posterior

probability to every hydrological parameter set as new taxi data become available.

Two components are critical for this Bayesian updating procedure. One is the prior probability and the other is the likelihood function. Regarding the prior probability, for their famous calibration model called generalized likelihood uncertainty estimation, Beven and Binley (1992) stated that all parameter combinations are considered equally probable before additional information is introduced. After the first update, the prior probability of each updating iteration can be

replaced by the posterior probability of the latest updating iteration. Likelihood, which is a measurement of how well a given model conforms to the observed taxi behavior, is not as easy to compute as the prior probability because the parameter set to be estimated is hydrology-related, whereas the observed evidence is taxi-related. Therefore, we must determine how to construct a taxi-based proxy whose probability is equal to the associated hydrological parameter and construct a function enabling the transformation from hydrological parameters to taxi-related proxies.

The proxy selected in this study was the time series of the no-taxi-passing probability. Figure 1 presents a generalized procedure for converting a rainfall time series into a time series of no-taxi-passing probabilities for each hydrological parameter. This procedure consists of three steps. First, a hydrological model is used to convert a rainfall time series into a hydrograph. Second, a runoff-disruption function that relates runoff to the probability that a road is blocked is used to transform the hydrograph into a time series of road disruption probabilities. Third, the taxi arrival rate is combined with the

time series of road disruption probabilities to derive a time series of no-taxi-passing probabilities. The hydrological model and taxi arrival rate are considered to be unique for every road and are invariable within a short period, whereas the runoff-disruption function is identical for all roads.

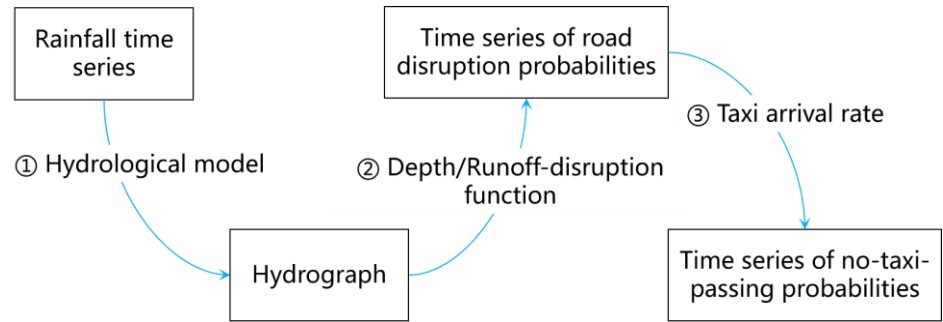

**Figure 1** Generalized procedure for converting a rainfall time series into a time series of no-taxi-passing probabilities.

Integrating this three-step process with the Bayesian equation enables us to compute the posterior probability of a parameter set based on taxi data. For a specific road, suppose there are $N$ hydrological parameter sets to be estimated. Because the runoff-disruption function and taxi arrival rate are assumed to be fixed for the road, we can construct a composite function converting the $i$th parameter set, which is denoted as $\boldsymbol{\theta}^{(i)}$, into a time series of no-taxi-passing probabilities, which is denoted

as $\boldsymbol{\Omega}^{(i)}$. Therefore, the probability of $\boldsymbol{\theta}^{(i)}$ being optimal is equal to the probability of $\boldsymbol{\Omega}^{(i)}$ being true, which can be expressed as follows:

$$P\big(\boldsymbol{\theta}^{(i)}\big) = P\big(\boldsymbol{\Omega}^{(i)}\big) \tag{1}$$

where $P\big(\boldsymbol{\theta}^{(i)}\big)$ and $P\big(\boldsymbol{\Omega}^{(i)}\big)$ are the prior probabilities of $\boldsymbol{\theta}^{(i)}$ and $\boldsymbol{\Omega}^{(i)}$ respectively. As taxi observations become available, $P\big(\boldsymbol{\theta}^{(i)}\big)$ (or $P\big(\boldsymbol{\Omega}^{(i)}\big)$) can be updated using the Bayes theorem as:

$$P\big(\boldsymbol{\theta}^{(i)}\big|X\big) = P\big(\boldsymbol{\Omega}^{(i)}\big|X\big) \propto P\big(\boldsymbol{\theta}^{(i)}\big)\mathcal{L}\big(X\big|\boldsymbol{\theta}^{(i)}\big) \tag{2}$$

where $X$ is the taxi observation, and $P\big(\boldsymbol{\theta}^{(i)}\big|X\big)$ and $P\big(\boldsymbol{\Omega}^{(i)}\big|X\big)$ are the posterior probabilities of $\boldsymbol{\theta}^{(i)}$ and $\boldsymbol{\Omega}^{(i)}$, respectively conditional on the taxi observation. $\mathcal{L}\big(X\big|\boldsymbol{\theta}^{(i)}\big)$ is the likelihood of $X$ given $\boldsymbol{\theta}^{(i)}$. The optimal parameter set is that which yields the $\Omega^{(i)}$ that most closely fits the observed taxi data.

Solving Eq. (2) involves the calculation of $P\big(\boldsymbol{\theta}^{(i)}\big)$ and $\mathcal{L}\big(X\big|\boldsymbol{\theta}^{(i)}\big)$. The derivation of $P\big(\boldsymbol{\theta}^{(i)}\big)$ depends on prior

knowledge regarding the parameter distribution, which is typically obtained using traditional hydrological methods. However, this prerequisite knowledge may not always be readily accessible based on limited data availability. In such cases, Beven and Binley (1992) suggested that any parameter set combination could be considered to be equally likely. This implies that the parameter set is drawn from a uniform distribution as follows:

$$P\big(\boldsymbol{\theta}^{(i)}\big) = 1/N \tag{3}$$

In this study, we compared the effects of two types of prior parameter distributions, namely a uniform distribution and a distribution derived from digital elevation model (DEM) data, on the resulting posterior distributions.

$\mathcal{L}\big(X\big|\boldsymbol{\theta}^{(i)}\big)$, which is a likelihood function, describes the joint probability of the observed taxi data $X$ as a function of the chosen $\theta^{(i)}$. Consider a rainfall event that is divided into $T$ 5 min intervals. From the taxi data, we can obtain a sequence of taxi-related observations, which are denoted as $X = \{x_1, x_2, \ldots, x_T\}$, where $x_t = 1$ if the observed road has at least one taxi

pass during the $t$th interval, and $x_t = 0$ otherwise. $\boldsymbol{\Omega}^{(i)} = \left\{\omega_1^{(i)}, \omega_2^{(i)}, \dots, \omega_T^{(i)}\right\}$ is also a $T$-dimensional vector, where $\omega_t^{(i)}$ is the no-taxi-passing probability in the $t$th interval with $\boldsymbol{\theta}^{(i)}$ as the parameter set. Note that $\boldsymbol{\Omega}^{(i)}$ is only determined by the chosen hydrological parameter and rainfall time series, and is not measured from observed data. Considering that the arrival of taxis is independent of time, $\mathcal{L}\left(\boldsymbol{X}\middle|\boldsymbol{\theta}^{(i)}\right)$ can be formulated as:

$$\mathcal{L}\left(\boldsymbol{X}\middle|\boldsymbol{\theta}^{(i)}\right) = \mathcal{L}\left(\boldsymbol{X}\middle|\boldsymbol{\Omega}^{(i)}\right) = \prod_{t=1}^{T}\left(1 - \omega_t^{(i)}\right)^{x_t}\left(\omega_t^{(i)}\right)^{1-x_t} \tag{4}$$

By substituting Eq. (4) into Eq. (2), the following equation can be obtained:

$$P\left(\boldsymbol{\theta}^{(i)}\middle|\boldsymbol{X}\right) \propto P\left(\boldsymbol{\theta}^{(i)}\right) \prod_{t=1}^{T}\left(1 - \omega_t^{(i)}\right)^{x_t}\left(\omega_t^{(i)}\right)^{1-x_t} \tag{5}$$

Equation (5) is the proposed Bayesian updating model for calibrating hydrological parameters based on taxi data, where $\boldsymbol{X}$ can be directly measured and $\omega_t^{(i)}$ is calculated through the three-step process illustrated in Fig. 1, which will be discussed in detail in the following section. Having selected an updating model, the optimal parameter for one period of observations may not be optimal for another period. Because the model may have continuing inputs of new taxi observations, the posterior probability for $\boldsymbol{\theta}^{(i)}$ should be updated as new evidence becomes available. For the second update, the posterior probability from the first observation becomes the prior probability for the second observation and the posterior probability for $\boldsymbol{\theta}^{(i)}$ is recursively updated as:

$$P\left(\boldsymbol{\theta}^{(i)}\middle|\boldsymbol{X_2}\right) \propto \mathcal{L}\left(\boldsymbol{X_2}\middle|\boldsymbol{\theta}^{(i)}\right)P\left(\boldsymbol{\theta}^{(i)}\middle|\boldsymbol{X_1}\right) \tag{6}$$

where $\boldsymbol{X_1}$ and $\boldsymbol{X_2}$ are the first and the second taxi observations.

## 2.2 Instantiation of the three-step procedure

Section 2.1 presented a generalized three-step procedure for converting a rainfall time series into a time series of no-taxi-passing probabilities. In this section, we specialize this process by integrating existing theories with our model. The three conceptualized steps illustrated in Fig. 1 were replaced with three concrete sub-models. First, a Soil Conservation Service (SCS) unit hydrograph was used to convert rainfall excess into a hydrograph of the target road. Second, an empirical runoff-disruption function based on data extracted from various experimental, observational, and modeling studies was applied to convert the hydrograph into a time series of road disruption probabilities. Third, a Poisson distribution representing the distribution of taxi arrival rate was combined with the road disruption probability time series to derive a no-taxi-passing probability time series.

## Step 1: Converting rainfall into runoff based on the SCS unit hydrograph

Not all rainfall produces runoff because soil storage can absorb a certain amount of rain. However, in urbanized areas, only a small proportion of rainfall infiltrates the soil or is retained on the land surface, leaving most rain to flow across urban surfaces and become direct runoff. The rainfall that becomes direct runoff is referred to as rainfall excess. The Natural

Resources Conservation Service (NRCS)[2] developed a method to estimate rainfall excess based on soil types and land uses

using the following curve number equation:

$$P_e = \begin{cases} (P_a - 0.2S)/(P_a + 0.8S) & P_a > 0.2S \\ 0 & P_a \le 0.2S \end{cases} \tag{7}$$

where $P_e$ is the accumulated rainfall excess in cm, $P_a$ is the accumulated rainfall in cm, and $S$ is the potential retention after runoff begins, which is defined as a function of the curve number as follows:

$$S = 2.54 \times (1000/CN - 10) \tag{8}$$

where $CN$ is the curve number. For urban and residential land, the curve number varies from 40 to 95 depending on the impervious area (Natural Resources Conservation Service, 2010a). Because prior knowledge on the $CN$ is unavailable, it was considered as a calibrated parameter in this study.

The rainfall excess derived using Eq. (7) was inputted into the unit hydrograph to derive the runoff. The unit hydrograph is a commonly used rainfall-runoff model that converts rainfall excess into a temporal distribution of direct

runoff. First proposed by Sherman in 1932, the unit hydrograph is defined as the hydrograph resulting from one unit of rainfall excess distributed uniformly over a catchment area. It assumes that rainfall is uniform over the catchment area and that runoff increases linearly with rainfall excess. Although these assumptions cannot be perfectly satisfied under most conditions, the results obtained from the unit hydrograph are generally acceptable for most practical cases. The model, originally designed for larger watersheds, has been found to be applicable to some catchment areas less than 5,000 m$^2$ in

size (Chow et al., 1988).

The unit hydrograph is only applicable to watershed areas where runoff data are measured. The paucity of runoff data motivated the development of the synthetic unit hydrograph (SUH) concept. The term "synthetic" in SUH refers to a unit hydrograph derived from watershed characteristics, rather than empirical rainfall-runoff relationships. In this study, we utilized the SCS unit hydrograph, which is a dimensionless SUH proposed by the NRCS. For the dimensionless SUH, the discharge

(i.e., $y$ axis) is expressed as the ratio of discharge $q$ to peak discharge $q_p$ and the time (i.e., $x$ axis) is expressed as the ratio of time $t$ to peak time $t_p$. Therefore, the SCS unit hydrograph is not exactly an SUH itself, but is a useful tool for constructing an SUH.

The shape of an SCS unit hydrograph is entirely determined by the peak rate factor. A standard value of 2.08 for the peak rate factor is recommended and commonly used by the NRCS (Fig. 2). To construct an SUH from an SCS unit hydrograph,

the $x$ axis of the SCS unit hydrograph is multiplied by $t_p$ and the $y$ axis is multiplied by $q_p$. The values of $q_p$ and $t_p$ are functions of the catchment area and time of concentration as follows:

$$t_p = 0.6t_c + D/2 \tag{9}$$

$$q_p = 2.08A/t_p \tag{10}$$

---

[2] The NRCS was originally called the US Soil Conservation Service.

where $t_C$ is the time of concentration in hours, $A$ is the catchment area in km$^2$, and $D$ is the duration of unit rainfall excess in hours, which was set to one-twelfth of an hour (i.e., 5 min) in this study. Notably, the catchment area and time of concentration are required to construct an SUH and they are the other two hydrological parameters that should be calibrated based on taxi data. Although numerous tools and theories have been developed for estimating catchment area and time of concentration, these two parameters are still prone to significant errors, particularly in urban areas, because of challenges in accurately delineating urban catchments (Huang and Jin, 2019; Li et al., 2020). Urban catchment delineation is more complex than natural catchment delineation. Urban catchments have spatially heterogeneous surface cover types, which change with city development and construction, and modify runoff parameters (Goodwin et al., 2009). High densities of residential and commercial buildings obstruct flow paths and alter flow directions of storm water runoff, complicating rainfall-runoff and overland flow processes in urban areas (Ji and Qiuwen, 2015). Additionally, accurate urban catchment delineation necessitates high-resolution DEMs, which are not always available. In many Chinese cities, high-resolution DEMs are considered confidential data and are generally inaccessible to non-governmental organizations. Based on these challenges, deriving accurate catchment area and time of concentration data in urban areas is difficult in Shenzhen.

For the sake of simplicity, the peak rate factor was not calibrated and was at 2.08, although some studies have indicated that it has a wide range from 0.43 for steep terrain to 2.58 for very flat terrain (Chow et al., 1988). After $t_C$ and $A$ were chosen, an SUH can be constructed and used it to convert rainfall excess into runoff by applying the discrete convolution equation. The detailed computation process of the discrete convolution equation can be found in most hydrological textbooks (e.g., Chow et al., (1988)), and will not be discussed here. The graphical workflow in Fig. 3 illustrates the transformation of rainfall time series data into a hydrograph for every parameter set.

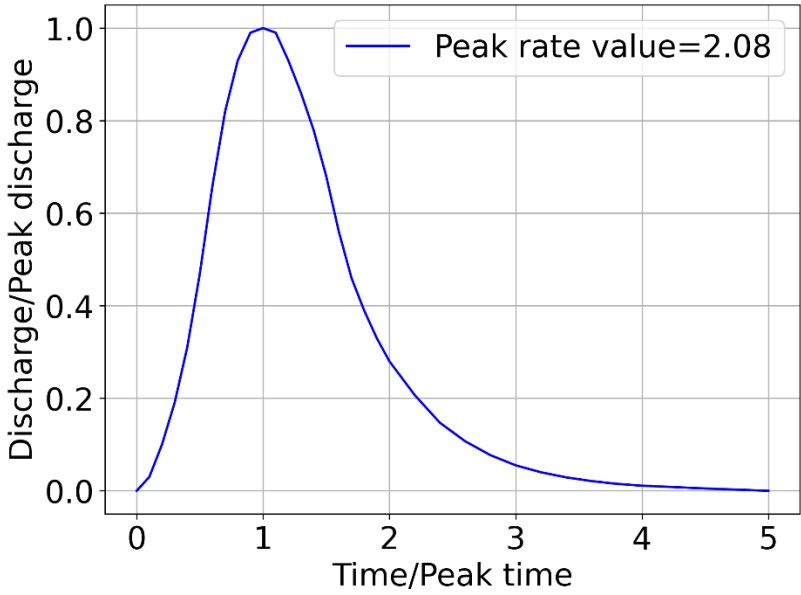

**Figure 2** Standard SCS unit hydrograph. Data provided by the NRCS (2007).

200

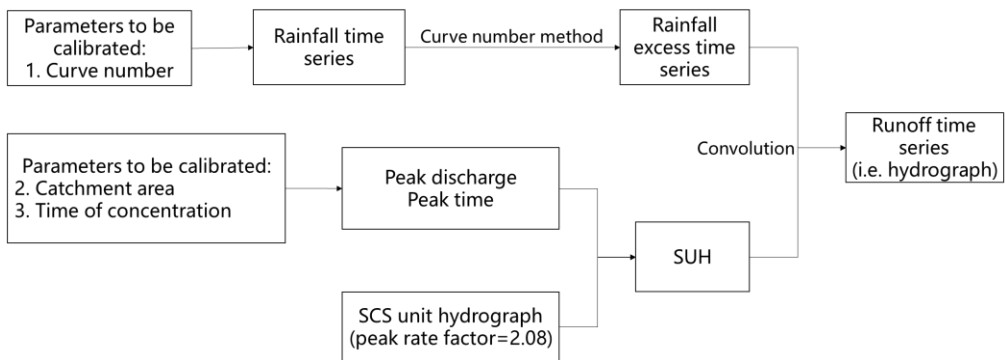

**Figure 3** Workflow of the SCS unit hydrograph for converting rainfall into runoff.

**Step 2: Derivation of the road disruption probability using the runoff-disruption function**

The goal of Step 2 is to convert the hydrograph generated in Step 1 into a time series of road disruption probabilities, or more specifically, the probability that a taxi driver chooses to turn their car when arriving at a flooded road. Most models in the literature assume that a road is either open or closed, which does not correspond to the empirical evidence that many drivers may take risks to drive on inundated roads. To transition from a binary view of a flooded road being considered "open" or "closed," Pregnolato et al. (2017) proposed the use of a curve that relates the depth of floodwater to a reduction in vehicle speed to indicate the probability of road disruption. This idea was soon adopted by Contreras-Jara et al. (2018) and Nieto et al. (2021).

A driver will turn around when he believes that the flow rate is too risky for their vehicle configuration. From this perspective, the road disruption probability is equal to the probability that vehicle performance is less than the flow rate perceived by a driver. However, it is difficult to quantify the factors that influence willingness of people to drive through a flooded roadway, and impossible to obtain the precise knowledge regarding all taxi-flood intersections. Alternatively, to ensure vehicle stability in flood flows, guidelines are typically recommended based on the limiting value of depth times velocity. Many researchers have conducted laboratory testing on the stability of different types of vehicle models exposed to different combinations of depth and velocity (Merz and Thieken, 2009; Shah et al., 2018). As suggested by Pregnolato et al. (2017), we constructed our runoff-disruption function by integrating data from the literature and authoritative guidelines. In this study, the road disruption probability was defined as the probability that the product of flow velocity and flow depth was greater than the stability limits extracted from the literature, which are listed in Table 1 and plotted in Fig. 4. The expression of the fitting curve is:

$$y = [1 + \exp(-16.6(x - 0.48)^2)]^{-1} \tag{11}$$

where $x$ is the product of flow velocity and flow depth, and $y$ is the disruption probability. According to Eq. (11), a road has a disruption probability of 50% when the product of flow velocity and flow depth is 0.47 m² s⁻¹ and is totally disrupted when the product is greater than 0.80 m² s⁻¹. By applying the fitting curve, we can easily convert the flood runoff into the disruption probability as follows:

$$P(Disrupt)_t^{(i)} = \left[1 + \exp\left(-16.6\left(q_t^{(i)}/W - 0.48\right)^2\right)\right]^{-1} \tag{12}$$

where $P(Disrupt)_t^{(i)}$ and $q_t^{(i)}$ are the road disruption probability and discharge in the $t$th interval derived from the hydrological model with the parameter set $\theta^{(i)}$, respectively, and $W$ is the road width.

230

**Table 1** Guidelines recommended in the existing literature.

| Reference | Vehicle type | Feature | Recommended limits for vehicle stability (m² s⁻¹) |
|---|---|---|---|
| Shah et al. (2018) | Volkswagen Scirocco | Flow direction =0° | velocity×depth<0.014 |
| Al-Qadami et al. (2022) | Perodua Viva | Ground clearance =0.18 m | velocity×depth<0.39 |
| Calculated according to Kramer et al. (2016) | VW Golf III | Not mention | velocity×depth<0.42 |
| Shand et al. (2016) | Large passenger | Ground clearance >0.12 m | velocity×depth<0.45 |
| Martínez-Gomariz et al. (2017) | Mini Cooper | Ground clearance =0.12 m | velocity×depth<0.49 |
| Martínez-Gomariz et al. (2017) | BMW i3 | Ground clearance =0.10 m | velocity×depth<0.49 |
| Martínez-Gomariz et al. (2017) | BMW 650 | Ground clearance =0.08 m | velocity×depth <0.50 |
| Martínez-Gomariz et al. (2017) | Mercedes GLA | Ground clearance =0.17 m | velocity×depth <0.59 |
| Moore and Power (2002) | All but very small cars | Not mention | velocity×depth <0.60 |
| Calculated according to Xia et al. (2014) | Honda Accord | Not mention | velocity×depth <0.65 |

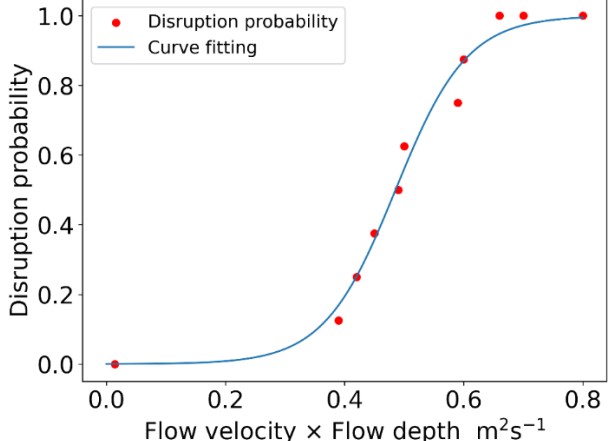

**Figure 4** Empirical runoff-disruption function derived from the existing literature.

**Step 3: Derivation of the time series of no-taxi-passing probabilities**

A road is considered to have no taxis passing in a fixed time interval if the road has no taxis arriving or if every taxi that arrives the road turns around. Therefore, the no-taxi-passing probability can be calculated using the following equation:

$$\omega_t^{(i)} = \sum_{n=0}^{\infty} P(Arrival\_taxi = n)_t \times \left(P(Disrupt)_t^{(i)}\right)^n \tag{13}$$

where $\omega_t^{(i)}$ is the no-taxi-passing probability in the $t$th interval, and $P(Arrival\_taxi = n)_t$ is the probability that $n$ taxis arrive at the road segment during the $t$th interval. Equation (13) indicates that if every taxi that arrives at the road segment makes a turn because of the flooded roadway, then the taxi volume on the road will be zero. In this study, $P(Arrival\_taxi = n)_t$ was assumed to follow the Poisson distribution:

$$P(Arrival\_taxi = n)_t = e^{-\lambda_t}\lambda_t^{\,n}/n! \tag{14}$$

where $\lambda_t$ is the average number of taxis arriving at the road during the $t$th interval. By substituting Eq. (14) in Eq. (13), we obtain:

$$\omega_t^{(i)} = \sum_{n=0}^{\infty} \left(e^{-\lambda_t}\lambda_t^{\,n}/n!\right) \times (P(Disrupt)_t^{(i)})^n \tag{15}$$

By applying $e^x = \sum_{n=0}^{\infty} x^n/n!$, Eq. (15) can be converted into:

$$\omega_t^{(i)} = e^{-\lambda_t} \sum_{n=0}^{\infty} \left(P(Disrupt)_t^{(i)}\lambda_t\right)^n/n! = \exp\left(\lambda_t\left(P(Disrupt)_t^{(i)} - 1\right)\right) \tag{16}$$

Equation (16) indicates that $\omega_t^{(i)}$ is entirely determined by $\lambda_t$ and $P(Disrupt)_t^{(i)}$. Because $P(Disrupt)_t^{(i)}$ is obtained from Step 2, what is left to determine is the value of $\lambda_t$. The value of $\lambda_t$ fluctuates according to the time of day, indicating higher taxi volume during congested periods and lower volume during non-congested periods. Therefore, we calculate $\lambda_t$ by averaging the taxi volume during the $t$th interval to account for time-of-day variations. It should be noted that as the intensity of rain increases, experienced taxi drivers will avoid flood-prone roads in advance, meaning that strictly speaking, $\lambda_t$ is a decreasing function of rainfall intensity. However, fitting the rainfall-$\lambda_t$ curve requires many taxi GPS trajectories to inspect the route choices of taxi drivers under heavy rain, which is outside the scope of this study. Therefore, we assumed that $\lambda_t$ was independent of rainfall.

Table 2 lists all the sub-models and parameters used in the three-step process. The core principle of the three-step process was to calculate the time series of no-taxi-passing probabilities, $\Omega^{(i)}$ given each parameter set $\theta^{(i)}$. Because the best choice of model is often data specific, it is likely that the model combination proposed in this study is not optimal for other scenarios. To apply the proposed calibration method in practice, the sub-models for the three-step process must be specified according to the available data, prior knowledge, and accuracy requirements.

**Table 2** Specific sub-models and parameters used in the three-step process.

| Purpose of each step | Specific model | Parameter | Source of parameters |
|---|---|---|---|
| | Curve number equation | 1. Curve number | Parameters to be calibrated |

| Step 1: Convert rainfall data into a hydrograph | SCS unit hydrograph | 2. Catchment area<br>3. Time of concentration | Parameters to be calibrated |
|---|---|---|---|
| | | 4. Peak rate factor | Existing literature |
| Step 2: Convert the hydrograph into a time series of disruption probabilities | Empirical runoff-disruption function | 5. Limit of product of flow velocity and depth | Existing literature |
| Step 3: Convert the time series of disruption probabilities into a time series of no-taxi probabilities | Taxi arrival rate following Poisson distribution | 6. Average taxi volumes in 5 min periods | Taxi GPS data |

## 3 Working example

The method outlined above was tested on the intersection of Xinzhou road and Hongli road in Shenzhen, which is recognized as a flood-prone point by the Water Authority of Shenzhen Municipality. Recall that the parameters to be calibrated are the curve number $CN$, catchment area $A$, and time of concentration $t_C$. The parameter spaces for $CN$, $A$, and $t_C$.

are determined by DEMs and other prior knowledge, which will be discussed in Section 4. Table 3 presents the details of the parameter sets to be calibrated, which form $8 \times 20 \times 30 = 4{,}800$ possible combinations. For ease of exposition, we assume that all parameters are uniformly distributed.

**Table 3** Detailed information on parameter sets to be calibrated.

| Parameter | Annotation | Minimum | Maximum | Incremental | Number of possible parameter values |
|---|---|---|---|---|---|
| Curve number | $CN$ | 40 | 75 | 5 | 8 |
| Catchment area | $A$ | 0.1 km$^2$ | 0.29 km$^2$ | 0.01 km$^2$ | 20 |
| Time of concentration | $t_C$ | 0.75 h | 3.2 h | 1/12 h | 30 |

Taxi GPS data collected during two storm events that occurred on May 9, 2015 and May 23, 2015 are used to calibrate

the parameter sets for the target intersection. Rainfall time series data and taxi observations during these two storms are presented in Fig. 5. Each taxi observation contains two time series: the time series of taxi volumes at 5 min intervals and time series of road statuses at 5 min intervals. These were derived from the taxi volumes with a value of one if the taxi volume was greater than zero and a value of zero if the taxi volume was zero.

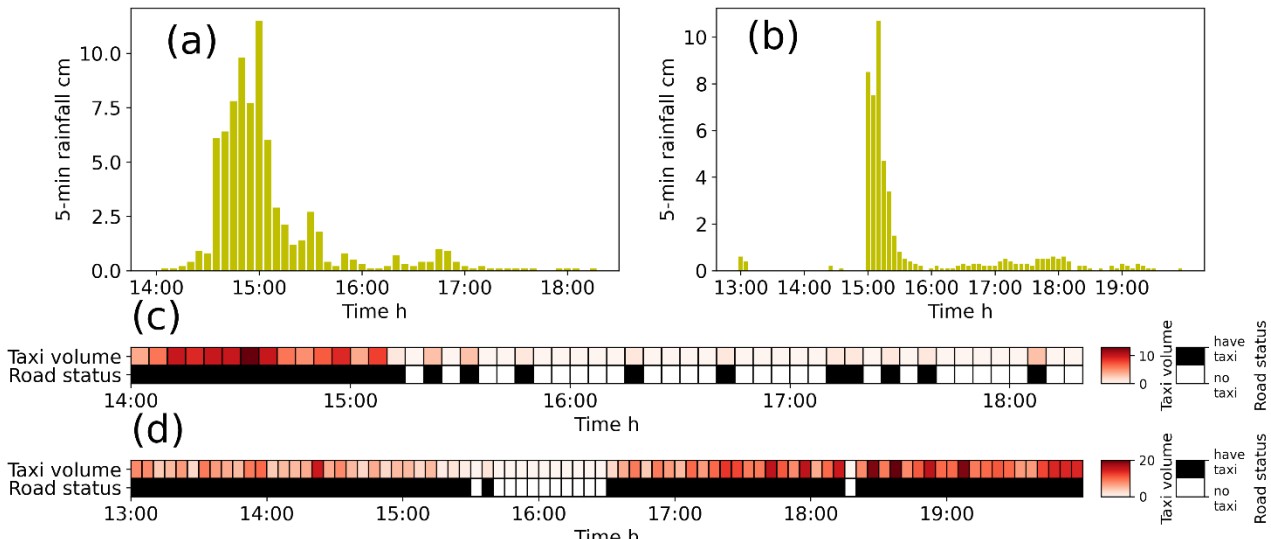

**Figure 5** Rainfall and taxi observations used to calibrate hydrological parameters: **(a)** Rainfall time series in 5 min intervals on May 9, 2015, **(b)** rainfall time series in 5 min intervals on May 23, 2015, **(c)** taxi observations on May 9, 2015, and **(d)** taxi observations on May 23, 2015.

Given the rainfall on May 9, 2015, we must calculate the time series of no-taxi-passing probabilities for each parameter combination. Because there are 4,800 parameter sets, we can derive 4,800 possible time series of no-taxi-passing probabilities. For simplicity, we only present the 3,120th parameter set (i.e., $CN = 65$, $A = 0.2$ km², and $t_C = 2.75$ h) as an example to demonstrate the working of the proposed method. According to the three-step process, the first step is to convert the original rainfall into rainfall excess using the curve number method given $CN = 65$ (Fig. 6(a)). Then, we calculated the peak discharge $q_p$ and peak time $t_p$ using Eqs. (9) and (10):

$$t_p = 0.6 \times 2.75 + \frac{1}{2 \times 12} \approx 1.69 \ h$$

$$q_p = 2.08 \times \frac{0.2}{1.69} \approx 0.24 \ \mathrm{m^3 s^{-1}}$$

The SUH was derived through multiplication by $t_p$ on the $x$ axis and $q_p$ on the $y$ axis of the standard SCS unit hydrograph (Fig. 6(b)). Next, the rainfall excess presented in Fig. 6(a) was combined with the derived SUH to obtain a hydrograph through convolution (Fig. 6(c)).

In the second step, the runoff was transformed into a time series of road disruption probabilities based on the runoff-disruption function (Fig. 6(d)). The runoff-disruption function takes the product of water depth and velocity (in units of m² s⁻¹) as inputs. Therefore, the original runoff (in units of m³ s⁻¹) derived in the first step should be divided by the road width before inputting it into the runoff-disruption function.

In the third step, the time series of road disruption probabilities (Fig.6(e)) was converted to no-taxi-passing probabilities using Eq. (16) (Fig. 6(f)). The average number of taxis during the flooding period is presented in Fig. (6f), and the derived time series of no-taxi-passing probabilities is presented in Fig. 6(g).

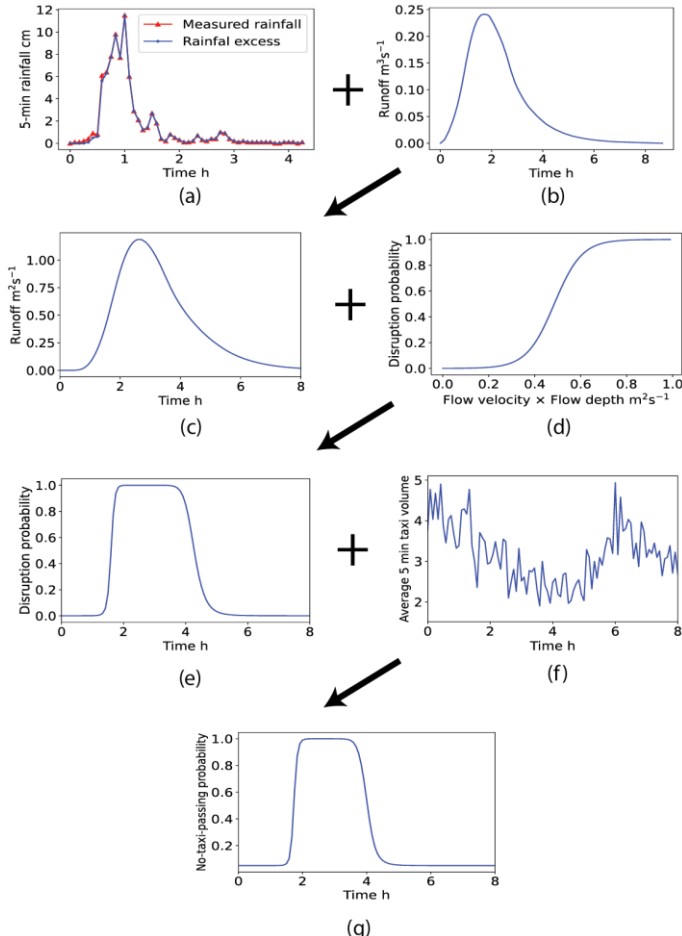

**Figure 6** Example transformation of a rainfall time series into no-taxi-passing probabilities using the three-step procedure for the 3,120th parameter set: **(a)** time series of rainfall and rainfall excess, **(b)** SUH constructed using the 3,120th parameter set, **(c)** derived runoff, **(d)** empirical runoff-disruption function, **(e)** derived time series of disruption probabilities, **(f)** average taxi volume in 5 min intervals, and **(g)** derived no-taxi-passing probabilities.

After the time series of no-taxi-passing probabilities for every parameter set were derived, the degree of belief that a given parameter set is optimal was calculated by integrating it with the taxi observations on May 9, 2015. According to Eq. (5), the posterior probability of the 3,120th parameter set is calculated as:

$$P\big(\theta^{(3120)}|X\big) \propto L\big(X|\theta^{(3120)}\big)P\big(\theta^{(3120)}\big) = 1/4800 \times \prod_{t=1}^{T}\big(1 - \omega_t^{(3120)}\big)^{x_t}\big(\omega_t^{(3120)}\big)^{1-x_t}$$

where $L\big(\theta^{(3120)}|X\big)$ is the posterior distribution of probabilities that the 3,120th parameter set is optimal conditional on $X$, which represents the taxi observations on May 9, 2015 presented in Fig. 5(c). $P\big(\theta^{(3120)}\big)$ is the prior probability of the 3,120th parameter set being optimal and its value is 1 / 4,800 because there are 4,800 possible combinations.

By following this process, we can calculate the posterior probabilities for every parameter set. Additionally, the posterior probability distribution of a parameter set can be updated using the taxi observations and rainfall data on May 23, 2015 as:

$$P(\boldsymbol{\theta}^{(i)}|\boldsymbol{X_2}) \propto \mathcal{L}(\boldsymbol{X_2}|\boldsymbol{\theta}^{(i)})P(\boldsymbol{\theta}^{(i)}|\boldsymbol{X_1})$$

where $P(\boldsymbol{\theta}^{(i)}|\boldsymbol{X_1})$ is the original posterior probability distribution calibrated based on the storm on May 9, 2015, and $P(\boldsymbol{\theta}^{(i)}|\boldsymbol{X_2})$ is the updated posterior distribution after the data of storm from May 23, 2015 are added. Fig. 7 illustrates the evolution of the probability distribution with the availability of additional taxi data. The first row in Fig. 7 represents the prior joint distribution of hydrological parameter sets, and the second and third rows represent the posterior distribution after each round of updating. The posterior distribution dominates the uniform prior distribution after the first update, and the distribution is refined slightly after the second update.

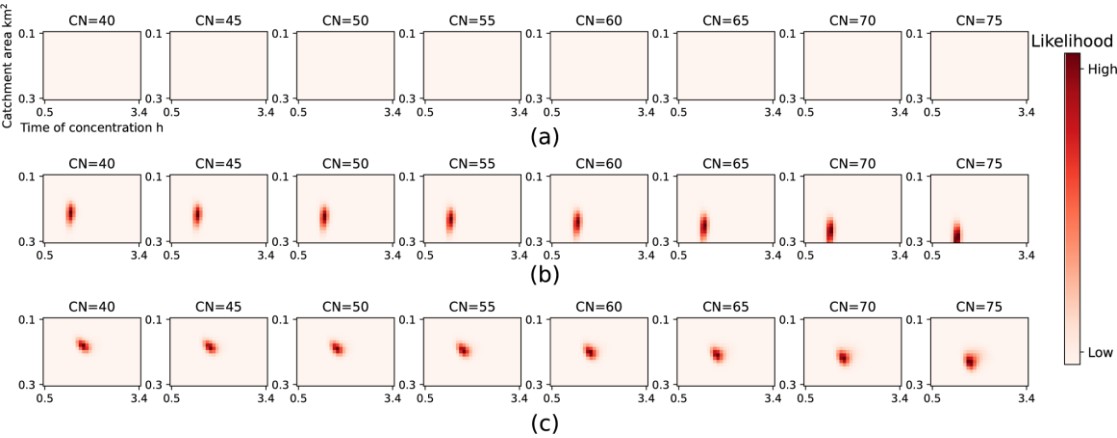

**Figure 7** Evolution of the posterior probability distribution of hydrological parameter sets: **(a)** Prior distribution before updating, **(b)** Posterior distribution after the first updating, and **(c)** Posterior distribution after the second updating.

## 4 Method validation

### 4.1 Data description

The proposed method was validated on flood-prone roads located in Shenzhen, China, which is a coastal city frequently hit by extreme storms during summer. To the best of our knowledge, Shenzhen is the only city that has shared runoff-related data with the public in China. Three data sources, namely taxi GPS data, rainfall data, and authoritative water level data, were used to validate our parameter calibration method. Hydrological parameters were calibrated using the first two data sources and the water level data acted as the ground truth to validate the proposed method. Taxi GPS data were anonymized and aggregated in 5 min intervals. Rainfall data, which were also collected in 5 min intervals, were measured at 115 gauging stations citywide and mapped to the road network throughout Shenzhen using the ordinary Kriging spatial interpolation algorithm. The water level data were only measured at certain flood-prone points with a dynamic sampling interval ranging from 5 min when the weather was rainy to 1 h when the weather was clear. The proposed calibration method was validated by

analyzing the hydrographs derived from the calibrated hydrological models against the authoritative water levels for 10 selected roads. Detailed information on the three data sources is provided in Table 4.


**Table 4** Detailed information on the three data sources.

| Item | Taxi GPS data[1] | Rainfall data[1] | Water level data[2] |
|---|---|---|---|
| Source | Transport Commission of Shenzhen Municipality | Meteorological Bureau of Shenzhen Municipality | Shenzhen Municipal Government Data Open Platform[1] |
| Record | Taxi volume of each road | 5 min accumulative rainfall | Water level |
| Data collection period | May 2015 | 2015 and 2019 | 2019 |
| Data collection interval | 5 min | 5 min | 5 min -1 h |
| Location | Citywide | 115 rainfall gaging stations | 171 flooding gaging sites |

[1] The complete taxi GPS data and rainfall data are not openly accessible owing to the requirements of data policy. To validate our research findings, we have uploaded the necessary data to Zenodo (Kong, 2022).

[2] Openly available at: https://opendata.sz.gov.cn/data/dataSet/toDataDetails/29200_01403147.

The two storm events on May 9, 2015 and May 23, 2015 were treated as calibration events and a storm on June 11, 2019,
was retained for testing. Clearly, there is a four year gap between the calibration data and validation data based on data availability. The hydrological environments of flood-prone roads may have changed during these years, which could render the parameters calibrated based on data from 2015 inaccurate for analysis in 2019. To reduce the validation error caused by this time gap, the roads to be validated should have been vulnerable to flooding in both 2015 and 2019 so that the hydrological parameters of these roads would have a higher chance of remaining unchanged. Therefore, a total of 10 flood-prone roads that
were labelled as such in both the List of 2015 Flood-prone Roads in Shenzhen (Water Authority of Shenzhen Municipality, 2015) and the List of 2019 Flood-prone Roads in Shenzhen (Water Authority of Shenzhen Municipality, 2019), were carefully selected (Fig. 8).

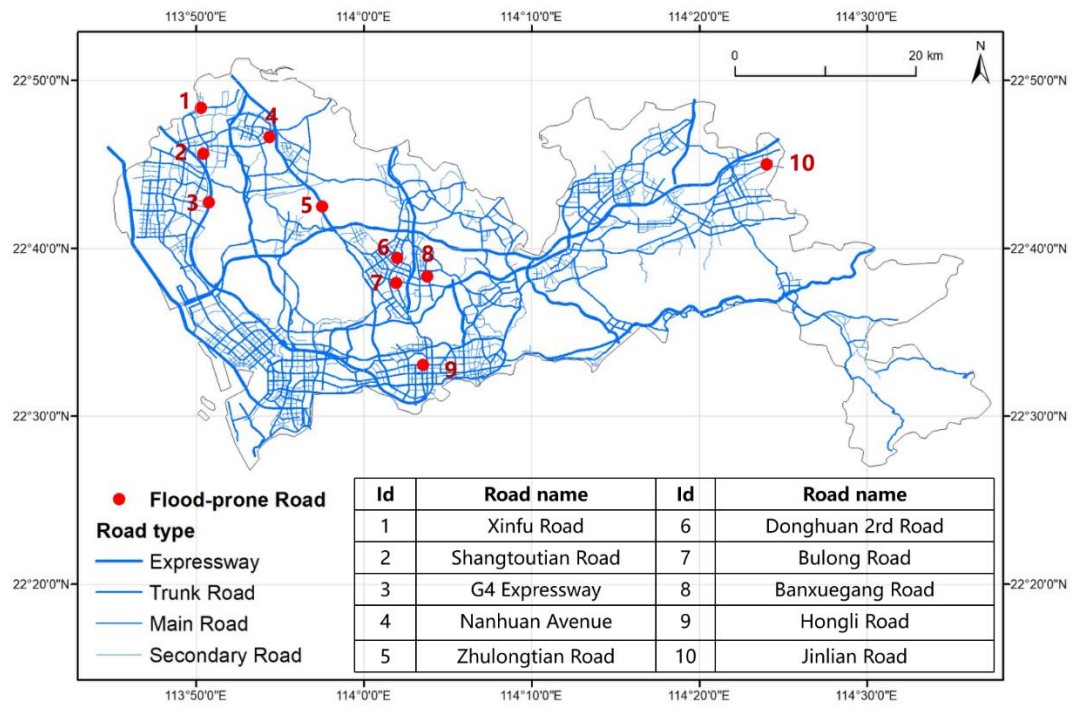

**Figure 8** Spatial distribution of 10 flood-prone roads in Shenzhen.

**4.2 Prior distributions of calibrated parameters**

We introduced two types of prior distributions to demonstrate the effects of prior distributions on calibrated parameters. The first prior distribution was determined based on prior knowledge and DEMs from Shenzhen, which were obtained from ASTER GDEM V3, which is a product of NASA and Japan's Ministry of Economy, Trade, and Industry (METI) (ASTER Global Digital Elevation Map, 2023). This global DEM covers the entire land surface of the earth with a 30 m resolution,

exhibiting notable improvements in horizontal and vertical accuracy while reducing anomalies compared to previous versions. We inputted the DEMs from Shenzhen into the hydrological software PCSWMM to delineate catchments and calculate the catchment area. Subsequently, we computed the time of concentration using the watershed lag method (Natural Resources Conservation Service, 2010b). As suggested by Zhang and Huang (2018), we used the average curve number for Shenzhen in 2015, which was assessed to be 60, as the estimated curve number for each road under validation.

We then constructed a discretized parameter space for the three parameters for each road as follows. For the curve number, we examined eight possible values centered on 60 with steps of five. For the catchment area, we considered 20 possible values centered on the estimated value with steps of 0.01 km². For the time of concentration, we considered 30 possible values centered on the estimated value with steps of 5 min. After constructing the parameter space for the parameters, we assigned a triangular prior distribution to each, which assumed the maximum probability at the estimated

value and linearly decreased to zero at the parameter space boundaries, as depicted in Fig. 9.

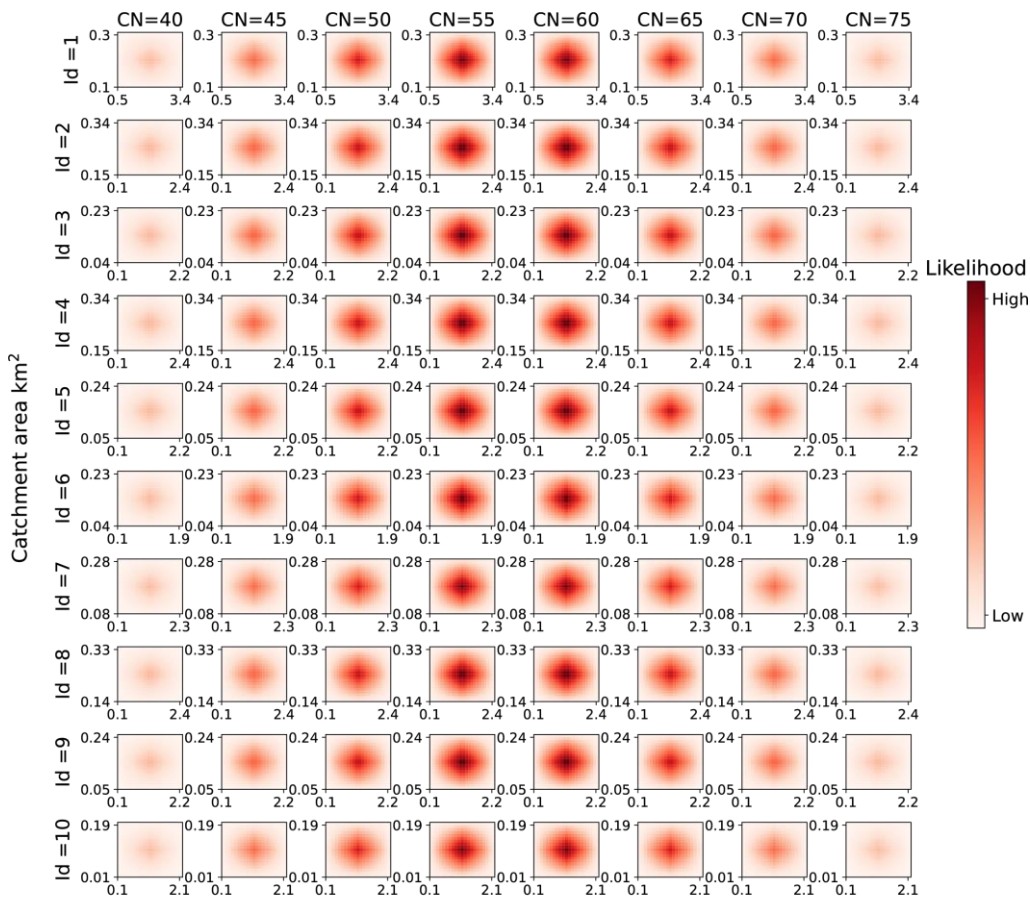

**Figure 9** Prior probability distributions of hydrological parameter sets based on DEMs and other prior knowledge for 10 flood-prone roads.

The second prior distribution assumed that the three parameters all follow uniform distributions. The parameter spaces for the second prior distribution were the same as those for the first. As a result, the joint probability of each parameter set was equal to $(1 / 20) \times (1 / 30) \times (1 / 8)$. To facilitate comparisons, we present the detailed information on the two types of prior distributions in Table 4.

**Table 4** Detailed information of the two types of prior distributions.

| Item | Prior probability distributions based on DEM and other prior knowledge | | | Uniform distributions | | |
|---|---|---|---|---|---|---|
| | Number of possible values | Parameter interval | Prior marginal distribution | Number of possible values | Parameter interval | Prior marginal distribution |

| | | | | | | |
|---|---|---|---|---|---|---|
| Curve number | 8 | 5 | Maximum probability at 60 and linearly reduces to zero at the parameter space boundaries | 8 | 5 | 1/8 for each possible value |
| Catchment area | 20 | 0.01 | Maximum probability at the estimated value and linearly reduces to zero at the parameter space boundaries | 20 | 0.01 | 1/20 for each possible value |
| Time of concentration | 30 | 1/12 | | 30 | 1/12 | 1/30 for each possible value |

### 4.3 Posterior distributions after calibration

We first calibrated the parameters based on the prior distributions calculated according to the DEMs and other prior knowledge. The resulting posterior distributions are presented in Fig. 10. Each row in Fig. 10 represents a different road, and each column represents a curve number. Each subplot presents the joint probability distribution of the catchment area and time of concentration for a given curve number. The color intensity in Fig. 10 represents the magnitude of the probabilities. Following two iterations of updating, the posterior probability distributions for both the catchment area and time of

concentration converge around the optimal parameter sets for most flood-prone roads. This demonstrates that incorporating taxi observations significantly reduces the uncertainty associated with catchment area and time of concentration. The probability typically achieves its maximum value when the curve number is either 55 or 60. Furthermore, each subplot contains a salient cluster with higher probability than other regions, suggesting that there may be multiple acceptable parameter sets.

Furthermore, the optimal catchment area under a given curve number decreases as the curve number increases, whereas the optimal time of concentration under a given curve number increases with the curve number. This is logical, because a higher curve number corresponds to increased rainfall excess under identical rainfall conditions, requiring a reduction in catchment area to maintain the runoff that best aligns with the taxi observations. Similarly, an increase in the time of concentration diminishes the peak runoff produced by the additional runoff generated by a higher curve number, thereby

preserving the optimal runoff status.

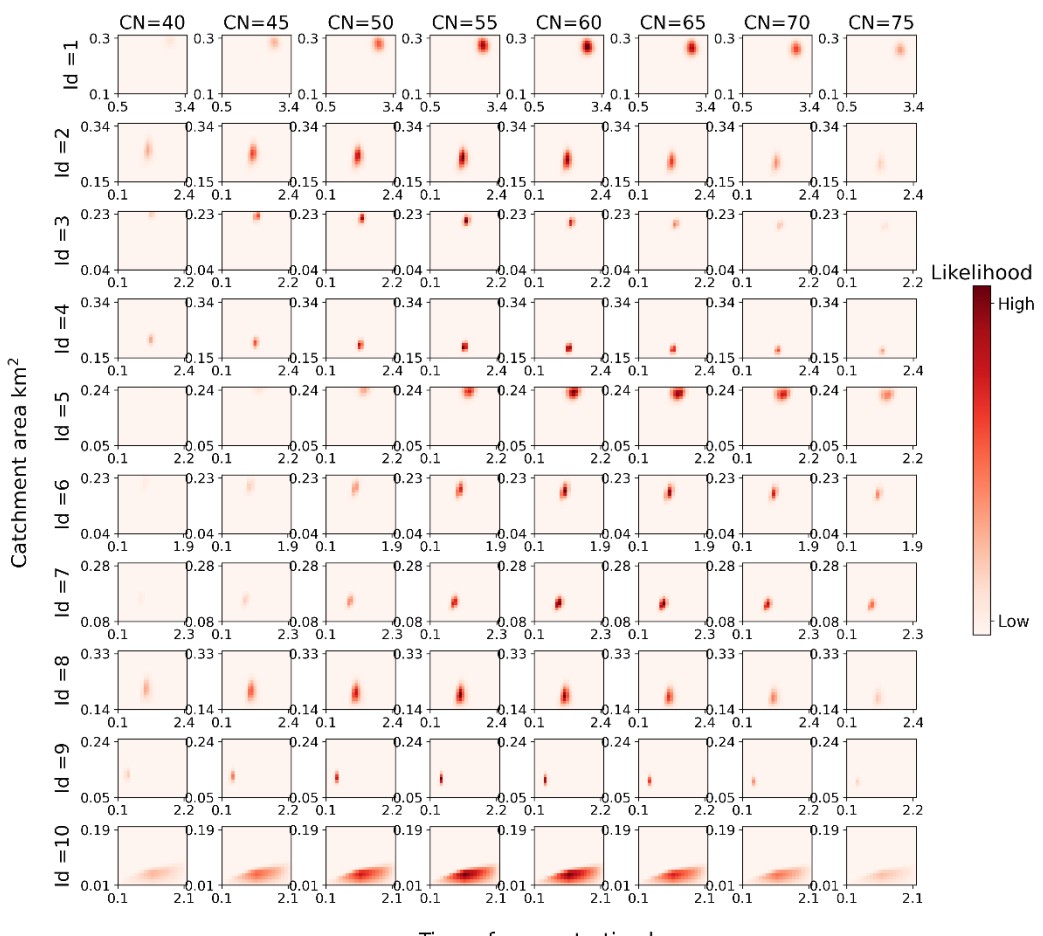

**Figure 10** Posterior probability distributions of hydrological parameter sets for 10 flood-prone roads after calibration. The prior probability distributions were derived from the DEMs and additional prior knowledge.

We also present the marginal distributions of the three parameters for 10 roads before and after calibration in Fig. 11. In Fig. 11, the marginal posterior distributions of the curve number appear relatively similar to the marginal prior distributions. It seems that the proposed method employing taxi data provides limited information regarding the distribution of curve numbers compared to the catchment area and time of concentration. This outcome may be a result of the range and discretization granularity of the parameter spaces. Catchment area and time of concentration encompass 20 and 30 possible values, respectively, whereas the curve number has only 8 potential values. The smaller parameter space of the curve number reduces the search space, and its impact on the no-taxi-passing probability is comparatively lower than that of the catchment area and time of concentration.

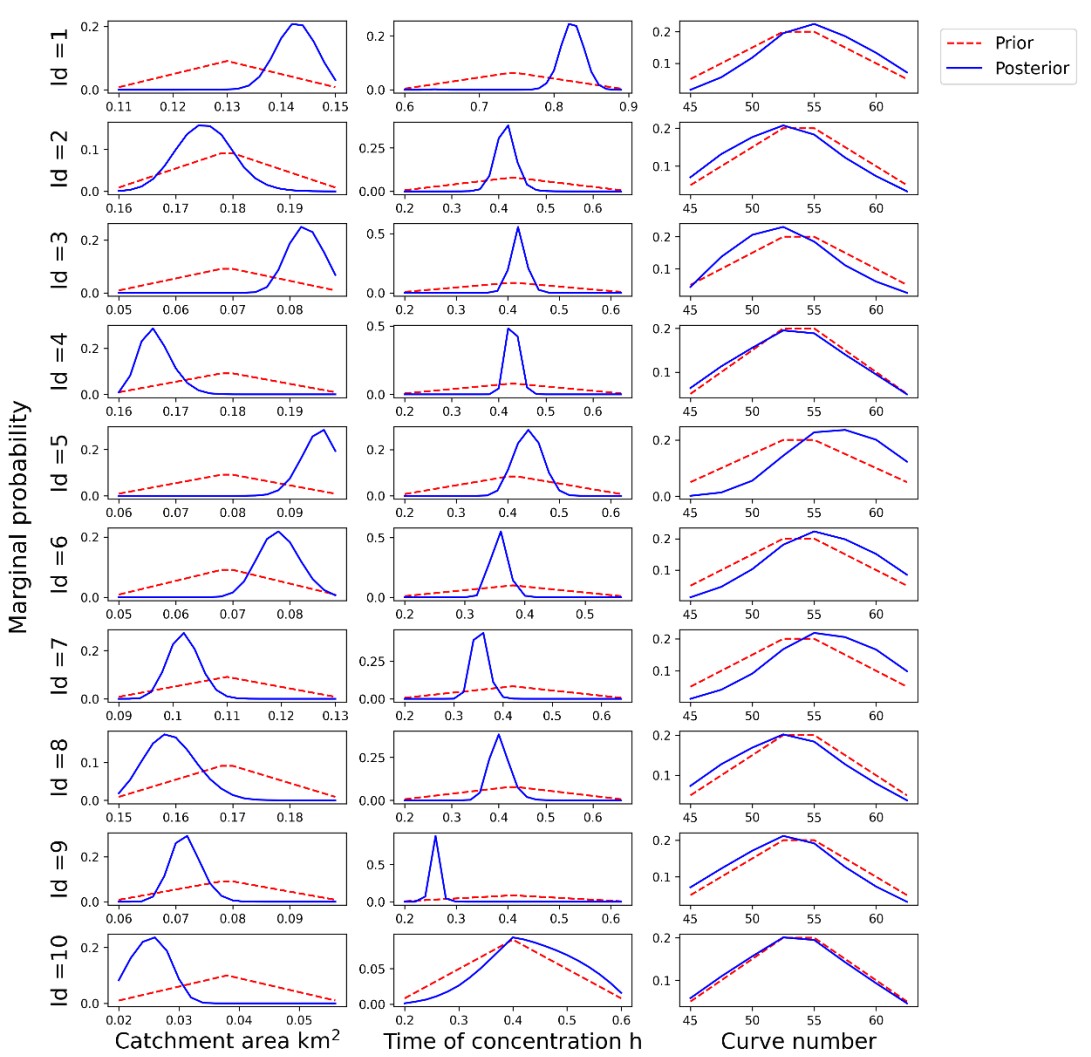

**Figure 11** Marginal prior and posterior probability distributions of the curve number for 10 flood-prone roads.

For example, for road ID = 6, the optimal parameter set consists of a catchment area of 0.19 km², time of concentration of 0.9 h, and curve number of 55. To investigate the effects of these parameters on the hydrograph and time series of no-taxi-passing probabilities, we held two parameters constant at their optimal values and observed the impact of changing the third parameter. Our findings are presented in Fig. 12. One can see that when the catchment area varies from 0.04 to 0.23 km², the maximum no-taxi-passing probability increases from 20% to 100% and the duration for which the no-taxi-passing probability exceeds 0.5 increases from 0.0 to 1.3 h. Similarly, when the time of concentration fluctuates from 0.1 to 1.9 h, the peak time of the no-taxi-passing probability varies from 0.5 to 1.8 h. In contrast, when the curve number varies from 40 to 75, the maximum no-taxi-passing probability is fixed at 100%, the duration for which the no-taxi-passing probability exceeds 0.5 extends from 1.1 to 1.3 h, and the peak time of the no-taxi-passing probability remains fixed at the 1.1 h. These

small fluctuations in the time series of no-taxi-passing probabilities are representative of why the distribution of curve numbers remains relatively stable after calibration compared to the catchment area and time of concentration.

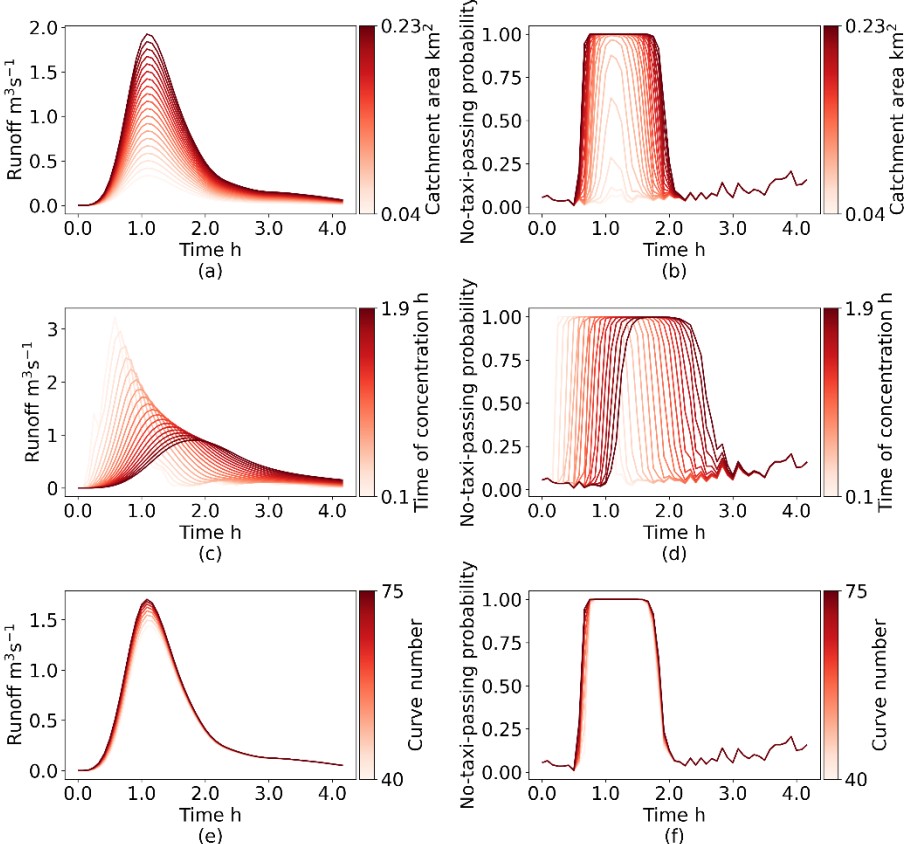


**Figure 12** Impacts of three parameters on the variation of the time series of runoff and no-taxi-passing probabilities: (a) catchment area conditional on runoff, (b) catchment area conditional on the no-taxi-passing probability, (c) time of concentration conditional on runoff, (d) time of concentration conditional on the no-taxi-passing probability, (e) curve number conditional on runoff, and (f) curve number conditional on the no-taxi-passing probability.

The posterior distributions calibrated based on the uniform prior distribution are presented in Fig. 13. When comparing two posterior distributions derived from two prior distributions, it is clear that the posterior distributions of the catchment area and time of concentration are very similar, indicating that the impact of prior distributions on these parameters rapidly diminishes after taxi-related knowledge is added. As stated by Beven and Binley (1992 pp: 286), "as soon as information is added in terms of comparisons between observed and predicted responses then, if this information has value, the distribution

of calculated likelihood values should dominate the uniform prior distribution when uncertainty estimates are recalculated."

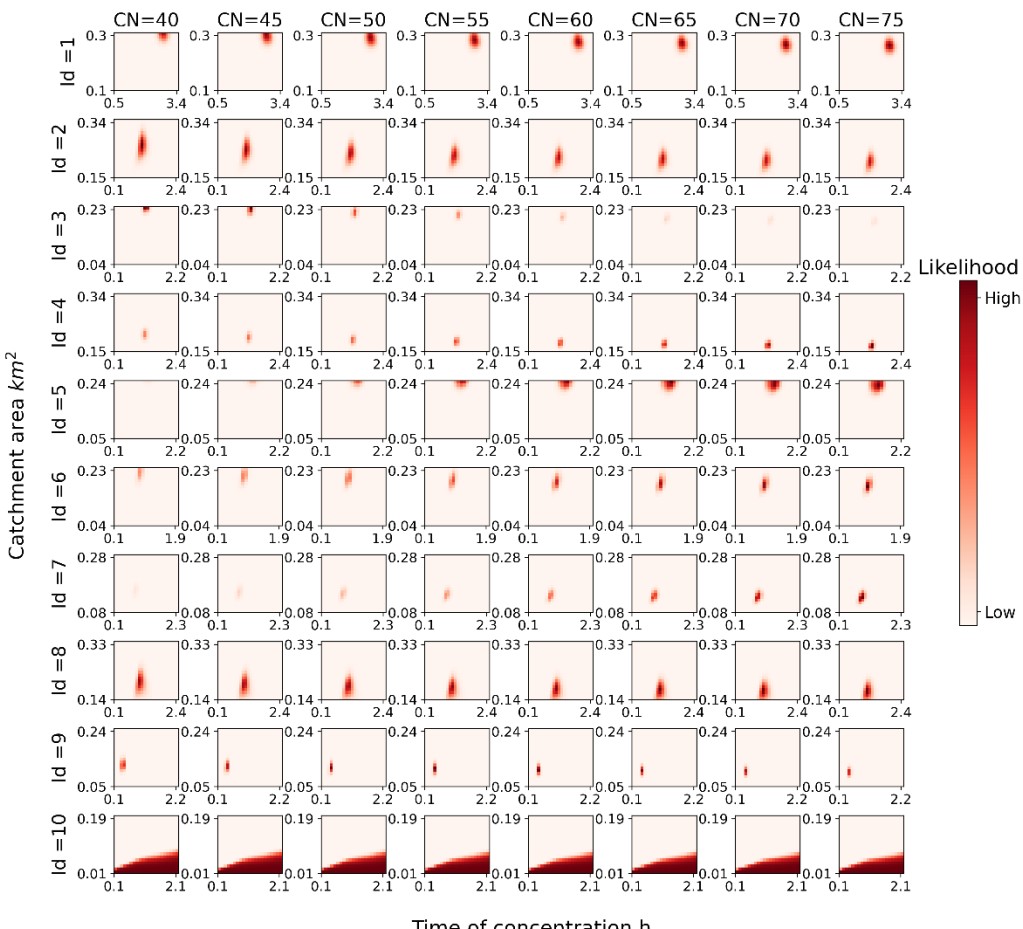

**Figure 13** Posterior probability distributions of hydrological parameter sets for 10 flood-prone roads after calibration. The prior probability distributions were derived from a uniform distribution.

### 4.4 Validation results

After the parameter sets were calibrated, they were combined with an SCS unit hydrograph to construct an SUH, which was combined with the rainfall data from June 11, 2019 to produce the predicted hydrograph. Because the posterior probability associated with each parameter set can be regarded as a fuzzy measure reflecting the degree of belief that the parameter set is true, the weighted runoff values for each parameter set were summed to calculate the final predicted runoff:

$$Q = \sum_{i=1}^{N} P\big(\theta^{(i)}|X\big)Q^{(i)} \tag{17}$$

Here, $Q$ is the final predicted runoff, $Q^{(i)}$ is the simulated runoff derived from the $i$th parameter set, and $P\big(\theta^{(i)}|X\big)$ is the posterior probability of the $i$th parameter set, which acts as a weight.

The output of the calibrated hydrological model is runoff (with units of $m^3 s^{-1}$), whereas the validation data are water level data (with units of $m$). Because the calibration data and validation data came from multiple sources and have different units, conventional error-based statistics such as the mean absolute error were not suitable for this study. The discharge of a
stream is rarely measured directly. Instead, streamflow is typically determined by converting measured water depth (i.e., water stage) into discharge through a rating curve, which provides a functional relationship between the water stage and discharge at a specified point (Le Coz et al., 2014). Inspired by the application of the rating curve, we validated our method by estimating the goodness of fit between the water level which was measured in the field and the corresponding runoff which was predicted based on the proposed calibration method. A higher goodness of fit indicates synchronous trends
between the runoff and water level, which indirectly demonstrates the feasibility of the proposed method.

       Because the posterior distributions derived from the two types of prior distributions were very similar, we only considered the posterior distribution calibrated based on prior distributions derived from DEMs and other prior knowledge for validation. Comparisons between the observed water depth and simulated runoff for 10 selected roads are presented in Fig. 14, and corresponding scatter plots are presented in Fig. 15. We use the Pearson correlation coefficient, which measures
the linear correlation between two variables, as a goodness of fit indicator. One can see that 8 of 10 roads are characterized by significant positive Pearson coefficients, indicating that the runoff and water have similar and consistent variation trends.

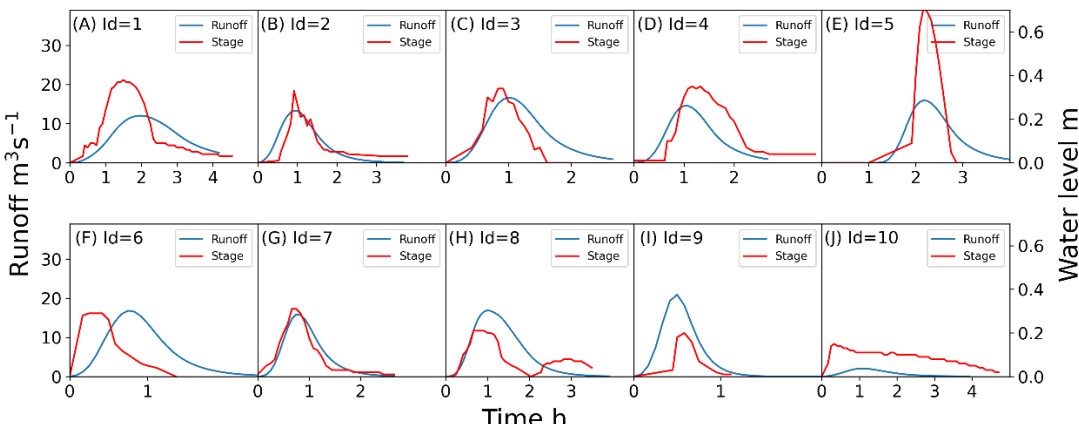

**Figure 14** Comparisons between the observed water depth and simulated runoff for roads 1 to10. The maximum value is 30
$m^3$ s$^{-1}$ on the left $y$ axis (i.e., runoff) and 0.6 m on the right $y$ axis (i.e., stage) for each subplot.

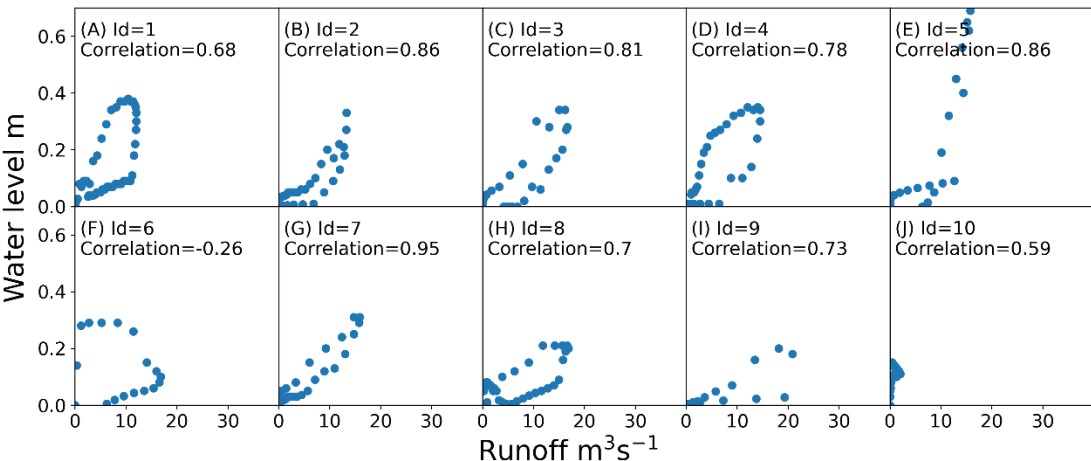

**Figure 15** Scatter plots of the observed water depth and the simulated runoff for roads 1-10.

It is noteworthy that goodness of fit simply describes the degree of correlation between the observed and simulated data, and may contain validation bias. As suggested by Legates and McCabe (1999), correlation-based statistics are insensitive to additive and proportional differences between simulations and observations. Therefore, the fitting of a rating curve is only a partial validation and the usefulness of the proposed calibration method requires further analysis.

## 5 Discussion

Four main points about the proposed calibration method are worthy of further discussion. The first is that although the presented validation results support the use of taxi GPS data to calibrate hydrological parameters for poorly gauged road networks, the proposed method is more applicable to roads that are frequently visited by taxis. Uncertainty increases as the taxi volume on a road decreases. A road is considered to be passable when at least one taxi GPS point is observed during a time interval, but we cannot assert that a road is disrupted when the taxi volume is zero. When a road with frequent taxi traffic is observed with no taxi GPS points during a storm, it is highly probable that the road is disrupted by flooding, which provides relatively reliable information for parameter calibration. Conversely, when a road with little taxi traffic has no taxi points during a storm, there is a relatively high likelihood that the road remains passable and is simply exhibiting its typical trend of no taxis. Therefore, the proposed calibration method becomes relatively unreliable when a no-taxi-passing period is no longer a good proxy for the disruption period on a road with sparse taxi data. To compensate for a shortage of taxi GPS data, additional data sources such as ride-hailing data and bus data should be incorporated in future work.

Second, the disruption of one road may cause cascading failures, where the disruption is rapidly propagated from the inundated road to adjacent non-inundated roads under the constraints of road connectivity. For a road that is disrupted, but not inundated by a storm, the implementation of the proposed calibration method may be subject to structural errors. Consider two connected roads called Road 1 and Road 2 that are both disrupted during a storm and have taxi volumes of

zero (Fig. 16). In this case, Road 1 is disrupted by the flooding, whereas Road 2 is only disrupted because it is connected to

Road 1. If taxi data are the only data used for calibration, then the posterior distributions of the hydrological parameters for Road 1 and Road 2 will be identical after calibration, because the sequences of taxi volume are identical for both roads. However, we know that the hydrological parameters of these two roads are not the same, because only one road is flooded. Just like we cannot simply treat the no-taxi-passing period as the disruption period, we cannot confuse the disruption period with the flooded period. In future work, an algorithm that facilitates distinguishing the flooding-induced disruption from

connectivity-induced disruption should be developed.

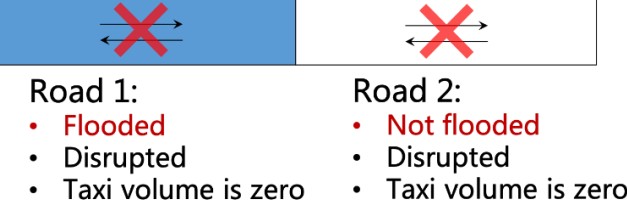

**Figure 16** Graphical representation highlighting the difference between the disruption period and flooded period

Third, the proposed three-step process, which consists of an SCS unit hydrograph, empirical runoff-disruption function, and Poisson distribution, is a realization of the generalized framework presented in Fig. 1. The sub-models used in the three-

step process can be flexibly replaced with other sub-models according to complexity requirements and data availability. For example, an alternative to the SCS unit hydrograph is the distributed hydrological model. Unlike the SCS unit hydrograph, the distributed hydrological model partitions a watershed into physically homogeneous units and captures the complex spatial variation induced by human activity in high resolution, which may be more applicable to urbanized environments, such as road networks. However, considering that some critical data such as road drainage data and land use data are

missing, as well as the extreme computational cost associated with the distributed hydrological model, we did not adopt this model in this study. Another assumption we made in this study is that the number of taxis arriving at a road follows a Poisson distribution. By conducting the Chi-square goodness of fit test, we found that the frequency distribution of taxi volumes adheres to a Poisson distribution for more than 50% of 5 min intervals for 7 of the 10 roads presented in Fig. 8, indicating that the Poisson model appears to be a suitable assumption. However, this hypothesis may not be universally

applicable, particularly in different urban contexts, where alternate distributions such as the Weibull distribution may provide a more accurate representation.

Fourth, it is imperative to acknowledge that the parameter values in this study were discretized, although hydrological model parameters are inherently continuous. This discretization approach could result in the omission of optimal solutions, particularly when hydrological models exhibit sensitivity to these parameters. It is important to note that discretization is

neither a requisite nor a recommended strategy. Future research should address the optimization or posterior inference problem in a continuous parameter space based on established methods such as the Monte Carlo algorithm.

## 6 Conclusion

An urban flooding model requires various types of data for calibration. In this study, we proposed a Bayesian calibration framework for the hydrological parameters of a road network based on taxi GPS data. A three-step procedure consisting of a rainfall-runoff model, runoff-disruption function, and no-taxi-passing probability model enabled us to transform a given rainfall time series into a time series of no-taxi-passing probabilities for each parameter set, which is key to taxi-data-driven model calibration. The calculated no-taxi-passing probabilities, which acted as a proxy for the associated hydrological parameter sets, were compared to observed taxi data based on the Bayes equation to assess the posterior probability distributions of the hydrological parameter sets. Three parameters, namely the curve number, catchment area, and time of concentration, were calibrated. The proposed calibration method was instantiated by combining classical hydrological models with traffic flow models and was validated on 10 flood-prone roads in Shenzhen. The validation results indicate that the trends of runoff could be correctly predicted for eight roads, which demonstrates the potential of calibrating hydrological parameters based on taxi GPS data.

This study highlights the potential of integrating transportation-related data with hydrological theory for the transportation resilience improvement and flood risk management of road networks. We hope that our study can provide a flexible calibration framework for countries that have little runoff data, but rich taxi data. We acknowledge that the application of the proposed method is currently limited by the heterogeneous spatial distributions of taxis citywide and cascading effects of road inundation, but expect this to change with the increasing availability of vehicle data and continuous optimization of modeling approaches.

## Code and data availability

The data and code used to validate the proposed method are available on Zenodo (https://doi.org/10.5281/zenodo.7894921).

## Author contributions

JY conceptualized the article and collected field data. XK designed the methodology and was responsible for code compilation. KX plotted the figures and revised the manuscript. BD managed the implementation of research activities. SJ discussed results and contributed to method validation. XK wrote the final version of the article with contributions from all co-authors.

## Competing interests

The corresponding author declares that none of the authors have any competing interests.

## Acknowledgments

This research was supported by the National Key Research and Development Program of China (Grant number No. 2022YFC3303100).

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
