# Peer review of "A Bayesian updating framework for calibrating the hydrological parameters of road networks using taxi GPS data"

_Hydrology and Earth System Sciences, 2023_

## Author Comment (AC1)

**Summary**

This manuscript presented a framework for calibrating a hydrologic model based on taxi data. The concept is quite clever, in my opinion. There of course is a need to calibrate hydrologic models and, at the same time, a general lack of data needed to calibrate. Using taxi data for the calibration is a neat idea and I think the authors did a good job showing the reader the feasibility of this. Overall, I think the manuscript is clear, well written, and technically sound. There are a few items I think should be addressed before being accepted for publication.

**Response:**

We express our gratitude to the reviewer for their insightful comments and suggestions, which will substantially improve the quality of our manuscript. Following careful consideration, we have amended the manuscript in accordance with your valuable comments. Our responses to your comments are provided below.

**Major comments**

1 - why are you calibrating time of concentration and catchment area? These are parameters that I would not typically see calibrated. It seems like you could estimate catchment area from a DEM. Similarly, there are many methods for estimating time of concentration from catchment characteristics. Because these two parameters are relatively reasonable to calculate/estimate, I'd like to understand the author's reasoning for calibrating them.

**Response:**

Thanks for your question. Although numerous tools and theories have been developed for estimating catchment area and time of concentration, these two parameters are still prone to significant errors, particularly in urban areas, due to challenges in accurately delineating urban catchments. First, urban catchment delineation is more complex than natural catchment delineation. Urban catchments have spatially heterogeneous surface cover types, which change with city development and construction, subsequently affecting runoff parameters (Goodwin et al., 2009). Unlike natural catchment, it is also difficult to identify explicit urban drainage systems and road slope directly from the topographic relief of the urban region. Furthermore, high densities of residential and commercial buildings obstruct flow paths and alter flow directions of stormwater runoff, complicating rainfall-runoff and overland flow processes in urban areas (Ji & Qiuwen, 2015).

Second, accurate urban catchment delineation necessitates high-resolution Digital Elevation Model (DEM), which are not always available in many regions. Oksanen and Sarjakoski (2005) demonstrated that automatic catchment delineation is highly sensitive to DEM errors, and uncertainty in DEMs determines the lower bound for catchment size that can be computed with sufficient accuracy. In many Chinese cities, high-resolution DEMs are considered confidential data and are generally inaccessible to non-governmental organizations. Consequently, using a low-resolution DEM may introduce substantial errors.

Due to these challenges, deriving accurate catchment area and time of concentration in urban areas is difficult. This study thus aims to provide an alternative method based on taxi GPS data to calibrate these parameters.

2 - why is a curve number of 85 used for every case? This seems pretty consequential since the CN could vary between catchments. Should this be a calibrated parameter?

**Response:**

Thank you for your suggestion. We acknowledge that fixing the curve number as 85 is not realistic as it is influenced by various factors in urban areas, such as impervious surface percentage and soil type. Therefore, we have revised the manuscript to include curve number as one of the parameters to be calibrated. In total, we calibrate three parameters: catchment area, time of concentration, and curve number.

Figure 1 presents the probability distributions of three parameters after calibration. Each row in Fig.1 represents a different road, and each column represents the curve number. Furthermore, each subplot shows the joint probability distribution of catchment area and time of concentration given the curve number. The depth of colour in Fig.1 represents the magnitude of probability. Following two iterations of updating, the posterior probability distribution for both catchment area and time of concentration converges around the optimal parameter sets for most flood-prone roads. This demonstrates that incorporating taxi observations has substantially narrowed the uncertainty associated with catchment area and time of concentration. The probability typically attains its maximum value when the curve number is either 55 or 60. Moreover, each subplot presents a salient cluster with higher probability than other regions, suggesting that there may be multiple parameter sets which can effectively represent the acceptable ones.

Furthermore, it has been observed that the optimal catchment area under a given curve number decreases as the curve number increases, while the optimal time of concentration under a given curve number rises in relation to the curve number. This is a logical observation, as a higher curve number corresponds to increased rainfall excess under identical rainfall conditions, requiring a reduction in catchment area to maintain the runoff that best aligns with the observed taxi-related road conditions. Likewise, the increase in time of concentration diminishes the peak runoff produced by the additional runoff generated by a higher curve number, thus preserving the optimal runoff status.

[Figure]

**Figure 1** Posterior probability distributions of hydrological parameter sets for 10 flood-prone roads after calibration. The prior probability distributions are derived from the DEM and additional prior knowledge.

We also plotted the marginal distributions of the three parameters for ten roads before and after calibration in Fig. 2. Upon examining Fig. 2, the marginal posterior distributions of the curve number appear relatively similar to the marginal prior distributions. It seems that the method employing taxi data provides limited information about the distribution of curve numbers after calibration compared to catchment area and time of concentration. This outcome may be attributed to the range and discretization granularity of the parameter spaces. Catchment area and time of concentration encompass 20 and 30 possible values, respectively, while the curve number includes only 8 potential values. The smaller parameter space of the curve number reduces the search space, and its impact on the no-taxi-passing probability is comparatively lower than that of catchment area and time of concentration.

[Figure]

**Figure 2** Marginal prior and posterior probability distribution of curve number for 10 flood-prone roads.

For instance, for road ID=6, the optimal parameter set maximizing the no-taxi-passing probability consists of a catchment area of 0.19 km², a time of concentration of 0.9 hour, and a curve number of 55. To investigate the effects of these parameters on the hydrograph and the time series of no-taxi-passing probability, we hold two parameters constant at their optimal values and observe the impact as the third parameter varied. Our findings, illustrated in Fig.3, demonstrate that when the catchment area varies from 0.04 km² to 0.23 km², the maximum no-taxi-passing probability ranges from 20% to 100%, and the duration of no-taxi-passing probability exceeding 0.5 hour extends from 0.0 to 1.3 hours. Similarly, when the time of concentration fluctuates from 0.1 to 1.9 hour, the peak time of no-taxi-passing probability spans from 0.5 to 1.8 hour. In contrast, when the curve number varies from 40 to 75, the maximum no-taxi-passing probability is fixed at 100%, the duration of no-taxi-passing probability excessing 0.5 hour extends from 1.1 to 1.3 hours, and the peak time of no-taxi-passing probability remains fixed at the 1.1 hour. The smaller fluctuations in the time series of no-taxi-passing probability interpret why the distribution of curve number remains relatively stable after calibration compared to catchment area and time of concentration.

[Figure]

**Figure 3** Impacts of three parameters on the variation of time series of runoff and no-taxi-passing probability. (a) Catchment area on the runoff. (b) Catchment area on the no-taxi-passing probability. (c) Time of concentration on the runoff. (d) Time of concentration on the no-taxi-passing probability (e) Curve number on the runoff. (f) Curve number on the no-taxi-passing probability

3 - Assuming it is reasonable to calibrate the catchment area and time of concentration, I question whether it's reasonable to have uniform priors for those parameters. Maybe you can't exactly know what the area of a catchment is going to be, but would you have enough of a guess to make a reasonable prior distribution? I'm guessing you'd know if a road segment has a relatively large or small catchment. Knowing this, it doesn't seem right to keep the prior distribution uniform.

**Response:**

In the original manuscript, we utilized only uniform priors for all parameters, leading to the inadequate use of prior knowledge, such as topography. In the revised manuscript, we introduce two types of prior distributions to demonstrate the effects of these distributions on calibrated parameters. The first prior distribution is determined based on the DEM of Shenzhen, obtained from ASTER GDEM V3, a product of NASA and Japan's Ministry of Economy, Trade, and Industry (METI) (Jet Propulsion Laboratory, 2019). This global digital elevation dataset covers the entire land surface of the earth with a 30-meter resolution and exhibits significant improvements in horizontal and vertical accuracy while reducing anomalies compared to previous versions. We

input the DEM of Shenzhen into the hydrological software PCSWMM to delineate the catchment and calculate the catchment area. Subsequently, we compute the time of concentration using the watershed lag method (Natural Resources Conservation Service, 2010). According to Zhang and Huang (2018), the average curve number for Shenzhen in 2015 was 60, which we adopt as the estimated curve number for the 10 roads.

We construct the discretized parameter space for each road's three parameters as follows: for the catchment area, we consider 20 possible values centered on the estimated value with 0.01 km² intervals; for the time of concentration, we explore 30 possible values centered on the estimated value with 0.1 hour intervals; and for the curve number, we examine 8 possible values centered on 60 with intervals of 5. After constructing the parameter space for three parameters, we assign a triangular prior distribution to each, which assumes maximum probability at the estimated value and linearly reduces to zero at the parameter space boundaries, as depicted in Fig.4.

[Figure]

**Figure 4** Prior probability distributions of hydrological parameter sets based on DEM and other prior knowledge for 10 flood-prone roads.

The second prior distribution assumes that the three parameters follow uniform distributions. The parameter space of the second prior distribution is the same as the first one. As a result, the joint probability of each parameter set equals $1/20 \times 1/30 \times 1/8$. Figure 5 presents the posterior distributions calibrated based on the uniform prior distribution. By comparing two posterior distributions derived from two prior distributions, it is evident that the posterior distributions of catchment area and time of concentration are close to each other, indicating that the impact of prior distributions on these parameters rapidly diminishes after taxi-related knowledge is added.

As stated by Beven and Binley (1992 pp: 286), "as soon as information is added in terms of comparisons between observed and predicted responses then, if this information has value, the distribution of calculated likelihood values should dominate the uniform prior distribution when uncertainty estimates are recalculated."

[Figure]

**Figure 5** Posterior probability distributions of hydrological parameter sets for 10 flood-prone roads after calibration. The prior probability distributions are derived from the uniform distribution.

4 - It may be that I didn't understand correctly, but how did you account for time of day/ day of week when considering whether or not a taxi would be passing? Or did you? For example, let's say that at a given roadway segment, there is a day and time of the week that there are hardly any taxis. Can you take that into account in your calibration scheme so that a lack of taxis then does not suggest to the model that the roadway is flooded?

**Response:**

We appreciate your valuable suggestion. In the previous version of our manuscript, we did not account for variations in taxi volume concerning the time-of-day or day-of-week. We assumed that the average number of taxis arriving on the road was constant, and the no-taxi-passing probability is given by:

$$\omega_t^{(i)} = e^{-\lambda} \sum_{n=0}^{\infty} \left( P(Disrupt)_t^{(i)} \lambda \right)^n / n! = \exp \left( \lambda \left( P(Disrupt)_t^{(i)} - 1 \right) \right)$$

where $\lambda$ is the average taxi volume per 5 min interval, calculated by averaging all 5 min taxi volumes using historical taxi GPS data for a specific road.

However, the value of $\lambda$ fluctuates according to the time of day, exhibiting higher taxi volume during congested periods and lower volume during non-congested periods. In the revised version, we incorporated the time-of-day variation in taxi volume when computing the no-taxi-passing probability:

$$\omega_t^{(i)} = e^{-\lambda} \sum_{n=0}^{\infty} \left( P(Disrupt)_t^{(i)} \lambda_t \right)^n / n! = \exp\left( \lambda_t \left( P(Disrupt)_t^{(i)} - 1 \right) \right)$$

where $\lambda_t$ is the 5 min taxi volume during the $t$th period, calculated by averaging the taxi volume of the $t$th period from May 1, 2015, to May 31, 2015. Compared with $\lambda$, $\lambda_t$ has smaller deviance because it excludes more non-flooding factors.

**Minor comments**

- Figure 5 - does it make sense to have intermittent "have taxis" and "no taxi" times after a large rain event? I guess I'm just wondering at graph (C) in particular where it looks like there is just one taxi between 16:15-16:20. Does that mean that one taxi is just really willing to risk it and drive through the water? If it's just one taxi, should it really be counted as "have taxi"?

**Response:**

This is a valid point. We also noticed that some drivers may take risks by driving through inundated roads, potentially resulting in intermittent "have-taxis" and "no-taxi" periods. We have examined the taxi volume between 16:15-16:20 and confirmed that one taxi drove through the water. Although the road appeared to be inundated and obstructed during this period, we would not categorize it as a theoretical "no-taxi" period. This is because our method determines the road's status ("have-taxis" or "no-taxi") based on the taxi volume, and the disruption period is inferred from the road's status. In other words, we predict the flood period according to the road's status, rather than vice versa. Furthermore, establishing an explicit rule to define "no-taxi" periods may cause confusion, as it implies that we have already observed the field data of flooding and constructed the rule based on it.

- Table 4 - if you had 171 flood gaging sites, why did you only pick 10 to test the model on? Why not test it on all 171?

**Response:**

The data used for parameter calibration were collected in 2015, while the data for method validation were collected in 2019, resulting in a four-year gap between the two datasets due to data availability. Furthermore, Shenzhen, as a coastal city, frequently experiences extreme storm events during summers. To mitigate flooding risks, the Shenzhen Municipal Government annually amends some flood-prone roads. As a result, the hydrological environment of certain roads may change over time, rendering parameters calibrated based on data in 2015 potentially inaccurate in 2019. To minimize validation errors caused by the time difference, we selected roads for validation that were vulnerable to flooding in both 2015 and 2019, increasing the likelihood that the hydrological parameters of these roads remained unchanged. Approximately 10 roads met this criterion. We will clarify this point in the revised manuscript.

- l355 - how did you make a rating curve for each road? How did you get the flow data to relate the stage data to?

**Response:**

The rating curve is usually determined by conducting field measurements and establish the relationship between the observed water level and the corresponding observed flow rate at a measuring point. In this study, however, we had no knowledge of empirical flow data for each road, thus we could not build a real rating curve. Instead, we establish a "rating curve" by plotting the water level which is field measured and the corresponding runoff which is predicted based on the proposed calibration method. If the derived "rating curve" is linearly related, indicating that the predicted runoff has the similar evolution trend of the observed water level, we thus assume that trends of runoff could be correctly predicted. To avoid confusion, we will not use the term "rating curve" to represent the relationship between the predicted runoff and the observed water level in the revised manuscript.

- Section 4.2 - I personally don't think you need this section. While it's interesting to see how you applied the framework, I don't think it is needed. I think it is enough to have described (section 2), illustrated (section 3), and validated (section 4) the method.

**Response:**

Thank you for your insightful suggestion regarding the section in question. After careful consideration, we agree that the mentioned section may not be necessary for our paper. As you pointed out, the method has been sufficiently described in Section 2, illustrated in Section 3, and validated in Section 4. In response to your suggestion, we will remove the section to streamline the manuscript and maintain focus on the key aspects of our research. We believe this revision will enhance the overall clarity and concision of our paper.

- L54: You might consider citing the following since they are related to this topic (full disclosure: I am an author on both):

- Sadler, J. M., Goodall, J. L., Morsy, M. M., & Spencer, K. (2018). Modeling urban coastal flood severity from crowd-sourced flood reports using Poisson regression and Random Forest. Journal of hydrology, 559, 43-55.

- Zahura, F. T., Goodall, J. L., Sadler, J. M., Shen, Y., Morsy, M. M., & Behl, M. (2020). Training machine learning surrogate models from a high-fidelity physics-based model: Application for real-time street-scale flood prediction in an urban coastal community. Water Resources Research, 56, e2019WR027038. https://doi.org/10.1029/2019WR027038

**Response:**

Thank you for your suggestion. We will cite these articles in the introduction section to enhance our review of existing research:

Citizens voluntarily or passively acting as human sensors generate georeferenced data to improve flood monitoring. Typical studies involve the use of crowdsourcing social media data (Brouwer & Eilander, 2017; Sadler et al., 2018; Zahura et al., 2020), mobile phone data (Yabe et al., 2018; Balistrocchi et al., 2020), and taxi GPS data (She et al., 2019; Kong et al., 2022).

- Figure 10: Could you explain why for some runoff values there is more than one level value? For empirically derived rating curves, each runoff value corresponds to only one water level.

**Response:**

As previously mentioned, the rating curve developed in our study relies on predicted runoff rather than observed runoff. Consequently, the temporal trends of the predicted results may not consistently align with those of the observed water levels. This discrepancy can result in one water level having two distinct runoff values. For instance, consider the road with ID=1 illustrated in Fig.6. When the water level reaches 0.27 m, the corresponding times are 1.1 hour and 1.2 hour. Due to the incongruity between the predicted runoff and observed water level, the runoff values at these two time points are 5.8 m³/s (Point A in Fig. 6) and 12 m³/s (Point B in Fig. 6), which accounts for the presence of two runoff values for a single water level.

[Figure]

**Figure 6** An example to show why some runoff values correspond to two level values.

**Editorial comments**

- l23 - suggest changing "metropolis" to plural "metropolises"
**Response:**
  Modified as suggested.

- l31 - suggest changing "false" to "incomplete" or "over-simplified"
**Response:**
  Modified to "incomplete" as suggested.

- l60 - suggest changing "critical" to "useful"
**Response:**
  Modified as suggested.

- l87 - I do not think you need to define a hydrograph. I think you can safely assume HESS readers will know what a hydrograph is.
**Response:**
  The definition of hydrograph is removed.

- l141 - "can absorb *a* light shower" (add "a")
**Response:**
  Modified as suggested.

- l154 - I suggest changing "converts rainfall excess to direct runoff" to "converts rainfall excess to a temporal distribution of direct runoff" or something like that to communicate that it is a distribution of runoff over time.

**Response:**

    Modified as suggested.

- l160 - "the paucity of runoff" instead of "the paucity of the runoff"

**Response:**

    Modified as suggested.

- l161 - "sparkled" is probably not the right word here. Maybe "sparked" or "motivated"

**Response:**

    Modified to "motivated" as suggested.

- l191 - "road" instead of "rood"

**Response:**

    Modified as suggested.

- l195 - "equals the probability" instead of "equals to the probability"

**Response:**

    Modified as suggested.

- l197 - suggest "impossible" instead of "difficult" because I think it is actually impossible to "obtain precise knowledge of all taxi-flooded intersections"

**Response:**

    Modified as suggested.

- Table 1: Is it correct to have the "/"s for Feature in several of the rows? If so, maybe you should define that means.

**Response:**

    Modified the "/" to "Not mention" in Table 1.

- l295 - suggest changing "a little bit" to "slightly" or something similar. "a little bit" is imprecise and colloquial

**Response:**

    Modified to "slightly" as suggested.

- l308 - "waterlogging" is not a term I typically hear. Do you mean something like "flood-prone?"

**Response:**

    Modified to "flood-prone" to enhance clarity.

- Figure 9 - is the x-axis "Time of Concentration?" If so, please change. I didn't know what "Time" meant.

**Response:**

Modified to "Time of Concentration" to enhance clarity.

- l396 - suggest replace "great" with "good"

**Response:**

Modified as suggested.

- l434 - suggest remove "great" to read "This study illustrates the potential ... "

**Response:**

Modified as suggested.

**Reference**

Beven, K., & Binley, A. (1992). The future of distributed models: Model calibration and uncertainty prediction. *Hydrological Processes*, *6*(3), 279–298. https://doi.org/10.1002/hyp.3360060305

Brouwer, T., & Eilander, D. (2017). Probabilistic flood extent estimates from social media flood observations. *Natural Hazards and Earth System Sciences*, *17*(5), 735–747.

Goodwin, N. R., Coops, N. C., Tooke, T. R., Christen, A., & Voogt, J. A. (2009). Characterizing urban surface cover and structure with airborne lidar technology. *Canadian Journal of Remote Sensing*, *35*(3), 297–309. https://doi.org/10.5589/m09-015

Jet Propulsion Laboratory. (2019). *ASTER Global Digital Elevation Map*. https://asterweb.jpl.nasa.gov/gdem.asp

Ji, S., & Qiuwen, Z. (2015). A GIS-based Subcatchments Division Approach for SWMM. *The Open Civil Engineering Journal*, *9*(1), 515–521. https://doi.org/10.2174/1874149501509010515

Natural Resources Conservation Service. (2010). *Hydrology National Engineering Handbook Chapter 15 Time of Concentration* (pp. 5–6). United States Department of Agriculture.

Oksanen, J., & Sarjakoski, T. (2005). Error propagation analysis of DEM-based drainage basin delineation. *International Journal of Remote Sensing*, *26*(14), 3085–3102. https://doi.org/10.1080/01431160500057947

Sadler, J. M., Goodall, J. L., Morsy, M. M., & Spencer, K. (2018). Modeling urban coastal flood severity from crowd-sourced flood reports using Poisson regression and Random Forest. *Journal of Hydrology*, *559*, 43–55. https://doi.org/10.1016/j.jhydrol.2018.01.044

Zahura, F. T., Goodall, J. L., Sadler, J. M., Shen, Y., Morsy, M. M., & Behl, M. (2020). Training Machine Learning Surrogate Models From a High-Fidelity Physics-Based Model: Application for Real-Time Street-Scale Flood Prediction in an Urban Coastal Community. *Water Resources Research*, *56*(10), e2019WR027038. https://doi.org/10.1029/2019WR027038

Zhang, T., & Huang, X. (2018). Monitoring of Urban Impervious Surfaces Using Time Series of High-Resolution Remote Sensing Images in Rapidly Urbanized Areas: A Case Study of Shenzhen. *IEEE Journal of Selected Topics in Applied Earth Observations and Remote Sensing*, *11*(8), 2692–2708. https://doi.org/10.1109/JSTARS.2018.2804440

---

## Author Comment (AC2)

**Summary**

The paper presents a novel approach for calibrating an urban rainfall-runoff model using taxi GPS data. This is an original idea that seems to have potential as demonstrated in this study. I also commend the authors for making available their data and code.

**Response:**

We thank the reviewer for the constructive comments to help us improve the manuscript. We are pleased that the manuscript aroused the reviewer's interest and are thankful for the positive feedback. Our responses to your comments are provided below. The data and code used to validate the method are available at Zenodo (https://doi.org/10.5281/zenodo.7894921).

**Major comments**

-What's the reason for modeling the taxi data as pass/no pass instead of directly modeling the number of taxis passing? The latter somehow seems more obvious since the original data are taxi counts, while your approach first requires converting taxi counts to 0/1 values, which introduces a potential loss of information. Please better justify this modeling choice.

**Response:**

We attempted to utilize taxi counts as an indicator of road status. However, establishing a relationship between precipitation or runoff and taxi count proved to be challenging. In situations where the road is not entirely disrupted by runoff, a stable quantitative relationship between taxi count and precipitation or runoff is hard to capture, as most taxis do not alter their travel routes

when the water level is not too high. Consequently, estimating $\mathscr{L}(X|\Omega^{(i)})$ becomes difficult

when $X$ represents taxi volume. In contrast, when the road is disrupted, the taxi volume should be zero by the definition of disruption, and typically greater than zero when the road is open,

simplifying the estimation of $\mathscr{L}(X|\Omega^{(i)})$. Overall, if the method calibrates parameters based on

the taxi count, it may select the optimal parameter that best corresponds to the observed taxi count rather than the road disruption status, thereby introducing unnecessary errors.

-An alternative approach would be to use the Poisson distribution to directly model the number of taxis passing (rather than arriving). Have you considered this? This would be more like a Poisson regression model, but perhaps leveraging your road disruption function to model lambda instead of the usual Poisson link function.

**Response:**

We appreciate your suggestion. As stated in our previous response, this approach requires establishing a numerical relationship between the road disruption function and lambda, which denotes the mean of the 5 min taxi volume. Since most taxis do not modify their travel routes during less intense runoff, the effect of rainfall on taxi volume becomes difficult to capture.

-Did you check whether the Poisson model for the number of taxis arriving at a road is a good assumption for your data?

**Response:**

In the revised manuscript, a Chi-square goodness-of-fit test is conducted to check whether the

frequency distribution adheres to a Poisson distribution. The squared differences between the observed and expected 5 min taxi frequencies, as predicted by the Poisson distribution, are computed to construct the Chi-square statistic. This statistic is then compared to the critical value corresponding to a significance level of 0.05. If the Chi-square statistic exceeds the critical value, the null hypothesis—that the probability distribution follows a Poisson distribution—is rejected; otherwise, the null hypothesis is accepted, suggesting the distribution may conform to a Poisson distribution.

Ten roads illustrated in Figure 8 of the manuscript were selected, and the frequency distribution of 5 min taxi volume for each period on each road was derived. Each frequency distribution consists of 31 samples, representing the taxi volume during a specific 5 min period collected from May 1, 2015, to May 31, 2015, for a specific road. The Chi-square goodness-of-fit test was applied to each frequency distribution, and the proportion of periods following a Poisson distribution for each road was calculated. The test results are presented in Table 1. According to these results, the frequency distribution of the 5 min taxi volume during a specific period adheres to the Poisson distribution for more than 50% of the periods in 7 out of the 10 roads. Consequently, the Poisson model appears to be a suitable assumption in this study.

**Table 1** Chi-square goodness-of-fit test of Poisson distribution for 10 roads.

| Road ID | 1 | 2 | 3 | 4 | 5 | 6 | 7 | 8 | 9 | 10 |
|---|---|---|---|---|---|---|---|---|---|---|
| Proportion of periods that follow Poisson distribution % | 12.5 | 95.8 | 79.2 | 97.5 | 85.8 | 27.5 | 96.7 | 48.3 | 73.3 | 95.8 |

-A limitation is that all variables are treated as discrete random variables whereas the hydrological model parameters are continuous. Why discretize the parameters?

**Response:**

The reason to discretize parameters stems from the challenges associated with solving optimal problems. While continuous parameters may yield more accurate estimations, it is often arduous to obtain an analytical or numerical solution for $\theta^{(i)}$ that maximizes $P(\theta^{(i)})\mathscr{L}(X|\theta^{(i)})$ from a continuous parameter space. For instance, finding the analytical solution which maximizes $P(\theta^{(i)})\mathscr{L}(X|\theta^{(i)})$ necessitates differentiating $\mathscr{L}(X|\theta^{(i)})$, which may not always be feasible.

Given a catchment area and a time of concentration, constructing a synthetic unit hydrograph (SUH) based on the SCS unit hydrograph is straightforward. However, determining these two parameters from a given SUH poses a significant challenge. Similarly, generating runoff by combining rainfall with the SUH through a convolution formula is easy, but deriving the SUH from rainfall and runoff is difficult. Chow et al. (1988) demonstrated deconvolution methods such as matrix calculations or linear programming to derive the SUH, but these approaches are complex and cannot provide explicit functions that input runoff and output SUH. To circumvent the backward parameter-solving process, we discretize continuous parameters and calculate $P(\theta^{(i)})\mathscr{L}(X|\theta^{(i)})$ for each parameter set using a forward calculation process, which is more convenient in this study. It is important to note that the exclusion of continuous parameters in this study is due to the complexity of differentiating the proposed three-step procedure, not an indication that they are inapplicable to other procedures.

**Response:**

The time-of-day variation may affect the taxi volume in a road. In the previous version of our manuscript, we did not account for variations in taxi volume concerning the time-of-day or day-of-week. We assumed that the average number of taxis arriving on the road was constant, and the no-taxi-passing probability is given by:

$$\omega_t^{(i)} = e^{-\lambda} \sum_{n=0}^{\infty} \left( P(Disrupt)_t^{(i)} \lambda \right)^n / n! = \exp\left( \lambda \left( P(Disrupt)_t^{(i)} - 1 \right) \right)$$

where $\lambda$ is the average taxi volume per 5 min interval, calculated by averaging all 5 min taxi volumes using historical taxi GPS data for a specific road.

However, the value of $\lambda$ fluctuates according to the time of day, exhibiting higher taxi volume during congested periods and lower volume during non-congested periods. In the revised version, we incorporated the time-of-day variation in taxi volume when computing the no-taxi-passing probability:

$$\omega_t^{(i)} = e^{-\lambda} \sum_{n=0}^{\infty} \left( P(Disrupt)_t^{(i)} \lambda_t \right)^n / n! = \exp\left( \lambda_t \left( P(Disrupt)_t^{(i)} - 1 \right) \right)$$

where $\lambda_t$ is the 5 min taxi volume during the $t$th period, calculated by averaging the taxi volume of the $t$th period from May 1, 2015, to May 31, 2015. Compared with $\lambda$, $\lambda_t$ has smaller deviance because it excludes more non-flooding factors.

**Response:**

Thank you for pointing this out. We acknowledge that fixing the curve number as 85 is not realistic as it is influenced by various factors in urban areas, such as impervious surface percentage and soil type. Therefore, we have revised the manuscript to include curve number as one of the parameters to be calibrated. In total, we calibrate three parameters: catchment area, time of concentration, and curve number.

Figure 1 presents the probability distributions of three parameters after calibration. Each row in Fig.1 represents a different road, and each column represents the curve number. Furthermore, each subplot shows the joint probability distributions of catchment area and time of concentration given the curve number. The depth of colour in Fig.1 represents the magnitude of probability. Following two iterations of updating, the posterior probability distribution for both catchment area and time of concentration converges around the optimal parameter sets for most flood-prone roads. This demonstrates that incorporating taxi observations has substantially narrowed the uncertainty associated with catchment area and time of concentration. The probability typically attains its maximum value when the curve number is either 55 or 60. Moreover, each subplot presents a salient cluster with higher probability than other regions, suggesting that there may be multiple parameter sets which can effectively represent the acceptable ones.

Additionally, it is observed that the optimal catchment area under a given curve number diminishes as the curve number rises, and the optimal time of concentration under a given curve

number increases in relation to the curve number. This is logical, as a higher curve number corresponds to increased rainfall excess given identical rainfall conditions, necessitating a decrease in catchment area to maintain the runoff that best aligns with the observed taxi-related road conditions. Similarly, the increase in time of concentration compensates for the additional runoff generated by a higher curve number, also preserving the optimal runoff status.

[Figure]

**Figure 1** Posterior probability distributions of hydrological parameter sets for 10 flood-prone roads after calibration. The prior probability distributions are derived from the DEM and additional prior knowledge.

In addition, we plotted the marginal distributions of three parameters for ten roads before and after calibration in Fig.2. Upon examining Fig.2, the marginal posterior distributions of curve number post-calibration appear relatively similar to the marginal prior distributions of curve number. It seems that the method employing taxi data offers limited information about the distribution of curve number after calibration in comparison to catchment area and time of concentration. This outcome may be ascribed to the range and discretization granularity of the parameter spaces. Catchment area and time of concentration encompass 20 and 30 possible values, respectively, whereas the curve number includes only eight potential values. The smaller parameter space of curve number diminishes the search space, and its impact on the no-taxi-passing probability is comparatively lower than that of catchment area and time of concentration.

[Figure]

**Figure 2** Marginal prior and posterior probability distribution of curve number for 10 flood-prone roads.

For instance, for road ID=6, the optimal parameter set maximizing the no-taxi-passing probability consists of a catchment area of 0.19 km², a time of concentration of 0.9 hour, and a curve number of 55. To investigate the effects of these parameters on the hydrograph and the time series of no-taxi-passing probability, we held two parameters constant at their optimal values and observed the impact as the third parameter varied. Our findings, illustrated in Fig.3, demonstrate that when the catchment area varies from 0.04 km² to 0.23 km², the maximum no-taxi-passing probability ranges from 20% to 100%, and the duration of no-taxi-passing probability exceeding 0.5 hour extends from 0.0 to 1.3 hours. Similarly, when the time of concentration fluctuates from 0.1 to 1.9 hour, the peak time of no-taxi-passing probability spans from 0.5 to 1.8 hour. In contrast, when the curve number varies from 40 to 75, the maximum no-taxi-passing probability is fixed at 100%, the duration of no-taxi-passing probability excessing 0.5 hour extends from 1.1 to 1.3 hours, and the peak time of no-taxi-passing probability remains fixed at the 1.1 hour. The smaller fluctuations in the time series of no-taxi-passing probability interpret why the distribution of curve number remains relatively stable after calibration compared to catchment area and time of concentration.

[Figure]

**Figure 3** Impacts of three parameters on the variation of time series of runoff and no-taxi-passing probability. (a) Catchment area on the runoff. (b) Catchment area on the no-taxi-passing probability. (c) Time of concentration on the runoff. (d) Time of concentration on the no-taxi-passing probability (e) Curve number on the runoff. (f) Curve number on the no-taxi-passing probability

-Section 2.1: a more common/general way is to write Bayes equation directly in terms of parameters theta, as in p(theta|X) \propto p(theta)*p(X|theta) or p(theta|X) \propto p(theta)*L(theta|X). The likelihood on the rhs of eq. 4 in the paper would then be written in terms of a function omega(theta) given by your eq. 16.

**Response:**

Thanks for your suggestion. The Bayes equation is rewritten as:

$$P(\theta^{(i)}\,|\,X)\;\propto\;P(\theta^{(i)})\mathscr{L}(X\,|\,\theta^{(i)})$$

**Editorial comments**

-eq. 11: please define x and y

**Response:**

The expression of the fitting curve is:

$$y=\left[1+\exp\left(-16.6\left(x-0.48\right)^{2}\right)\right]^{-1} \tag{11}$$

where $x$ is the product of flow velocity and flow depth, and $y$ is the disruption probability.

-L23: metropolis --> metropolises or metropolitan areas
**Response:**
Modified as suggested.

-L40: "calibrated on runoff data alone" - there are many studies that calibrate on other data as well
Response:
**Response:**
It is not rigorous to say that "No matter what kinds of methods, hydrological models are basically calibrated based on the runoff data alone." Thus, that sentence is removed.

-L47: ungaged vs ungauged: pick one spelling
**Response:**
Modified to "ungaged" as suggested.

-L83 (and other places): equals to --> equals
**Response:**
Modified as suggested.

-L90: arriving --> arrival
**Response:**
Modified as suggested.

-L99: does index i refer to road i?
**Response:**
No, the index $i$ refers to the $i$th parameter set.

-L132: instantization --> instantiation
**Response:**
Modified as suggested.

-suggest to proofread entire manuscript to fix issues with use of English
**Response:**
We thank the reviewer for pointing this out. As suggested, the manuscript is thoroughly proofread, and the grammar, clarity, and overall readability is also improved.

---

## Author Response (AR1)

**Response to Reviewer 1**

**Summary**

This manuscript presented a framework for calibrating a hydrologic model based on taxi data. The concept is quite clever, in my opinion. There of course is a need to calibrate hydrologic models and, at the same time, a general lack of data needed to calibrate. Using taxi data for the calibration is a neat idea and I think the authors did a good job showing the reader the feasibility of this. Overall, I think the manuscript is clear, well written, and technically sound. There are a few items I think should be addressed before being accepted for publication.

**Response:**

We express our gratitude to the reviewer for the insightful comments and suggestions, which substantially improved the quality of our manuscript. Following careful consideration, we have amended the manuscript in accordance with your valuable comments. Our responses to your comments are provided below.

**Major comments**

1 - why are you calibrating time of concentration and catchment area? These are parameters that I would not typically see calibrated. It seems like you could estimate catchment area from a DEM. Similarly, there are many methods for estimating time of concentration from catchment characteristics. Because these two parameters are relatively reasonable to calculate/estimate, I'd like to understand the author's reasoning for calibrating them.

**Response:**

Thanks for your question. Although numerous tools and theories have been developed for estimating catchment area and time of concentration, these two parameters are still prone to significant errors, particularly in urban areas, due to challenges in accurately delineating urban catchments. First, urban catchment delineation is more complex than natural catchment delineation. Urban catchments have spatially heterogeneous surface cover types, which change with city development and construction, and modify runoff parameters (Goodwin et al., 2009). Unlike natural catchment, it is also difficult to identify explicit urban drainage systems and road slope directly from the topographic relief of the urban region. Furthermore, high densities of residential and commercial buildings obstruct flow paths and alter flow directions of stormwater runoff, complicating rainfall-runoff and overland flow processes in urban areas (Ji & Qiuwen, 2015).

Second, accurate urban catchment delineation necessitates high-resolution Digital Elevation Model (DEM), which is not always available in many regions. Oksanen and Sarjakoski (2005) demonstrated that automatic catchment delineation is highly sensitive to DEM errors, and uncertainty in DEMs determines the lower bound for catchment size that can be computed with sufficient accuracy. In many Chinese cities, high-resolution DEMs are considered confidential data and are generally inaccessible to non-governmental organizations. Consequently, using a low-resolution DEM may introduce substantial errors.

Due to these challenges, deriving accurate catchment area and time of concentration in urban areas is difficult. This study thus aims to provide an alternative method based on taxi GPS data to

calibrate these parameters. We have clarified this in the revised manuscript (**Line 182**).

2 - why is a curve number of 85 used for every case? This seems pretty consequential since the CN could vary between catchments. Should this be a calibrated parameter?
**Response:**

Thank you for your suggestion. We acknowledge that fixing the curve number as 85 is not realistic as it is influenced by various factors in urban areas, such as impervious surface percentage and soil type. Therefore, we have revised the manuscript to include curve number as one of the parameters to be calibrated (**Line 375**). In total, we calibrate three parameters: catchment area, time of concentration, and curve number.

[revised manuscript text omitted]

We then constructed a discretized parameter space for the three parameters for each road as follows. For the curve number, we examined eight possible values centered on 60 with steps of five. For the catchment area, we considered 20 possible values centered on the estimated value with steps of 0.01 km2. For the time of concentration, we considered 30 possible values centered on the estimated value with steps of 5 min. After constructing the parameter space for the parameters, we assigned a triangular prior distribution to each, which assumed the maximum probability at the estimated value and linearly decreased to zero at the parameter space boundaries, as depicted in Fig. 4.

[Figure]

**Figure 4** Prior probability distributions of hydrological parameter sets based on DEMs and other prior knowledge for 10 flood-prone roads.

The second prior distribution assumed that the three parameters all follow uniform distributions. The parameter spaces for the second prior distribution were the same as those for the first. As a result, the joint probability of each parameter set was equal to $(1 / 20) \times (1 / 30) \times (1 / 8)$. Figure 5 presents the posterior distributions calibrated based on the uniform prior distribution. By comparing two posterior distributions derived from two prior distributions, it is evident that the posterior distributions of catchment area and time of concentration are close to each other,

indicating that the impact of prior distributions on these parameters rapidly diminishes after taxi-related knowledge is added. As stated by Beven and Binley (1992 pp: 286), "as soon as information is added in terms of comparisons between observed and predicted responses then, if this information has value, the distribution of calculated likelihood values should dominate the uniform prior distribution when uncertainty estimates are recalculated."

[Figure]

**Figure 5** Posterior probability distributions of hydrological parameter sets for 10 flood-prone roads after calibration. The prior probability distributions are derived from the uniform distribution.

4 - It may be that I didn't understand correctly, but how did you account for time of day/ day of week when considering whether or not a taxi would be passing? Or did you? For example, let's say that at a given roadway segment, there is a day and time of the week that there are hardly any taxis. Can you take that into account in your calibration scheme so that a lack of taxis then does not suggest to the model that the roadway is flooded?

**Response:**

We appreciate your valuable suggestion. In the previous version of our manuscript, we did not account for variations in taxi volume concerning the time-of-day or day-of-week. We assumed that the average number of taxis arriving on the road was constant, and the no-taxi-passing probability is given by:

$$\omega_t^{(i)} = e^{-\lambda_t} \sum_{n=0}^{\infty} \left( P(Disrupt)_t^{(i)} \lambda \right)^n / n! = \exp\left( \lambda \left( P(Disrupt)_t^{(i)} - 1 \right) \right) \tag{1}$$

where $\lambda$ is the average taxi volume in 5 min interval, calculated by averaging all 5 min taxi volumes using historical taxi GPS data for a specific road.

However, the value of $\lambda$ fluctuates according to the time of day, indicating higher taxi volume during congested periods and lower volume during non-congested periods. In the revised version (**Line 245**), we incorporated the time-of-day variation in taxi volume when computing the no-taxi-passing probability:

$$\omega_t^{(i)} = e^{-\lambda_t} \sum_{n=0}^{\infty} \left( P(Disrupt)_t^{(i)} \lambda_t \right)^n / n! = \exp\left( \lambda_t \left( P(Disrupt)_t^{(i)} - 1 \right) \right) \qquad (2)$$

where $\lambda_t$ is the average number of taxis arriving at the road during the $t$th interval. Compared with $\lambda$, $\lambda_t$ has smaller deviance because it excludes more non-flooding factors.

**Minor comments**

- Figure 5 - does it make sense to have intermittent "have taxis" and "no taxi" times after a large rain event? I guess I'm just wondering at graph (C) in particular where it looks like there is just one taxi between 16:15-16:20. Does that mean that one taxi is just really willing to risk it and drive through the water? If it's just one taxi, should it really be counted as "have taxi"?

**Response:**

This is a valid point. We also noticed that some drivers may take risks by driving through inundated roads, potentially resulting in intermittent "have-taxis" and "no-taxi" periods. We examined the taxi volume between 16:15-16:20 and confirmed that one taxi drove through the water. Although the road appeared to be inundated and obstructed during this period, we would not categorize it as a theoretical "no-taxi" period. This is because our method determines the road's status ("have-taxis" or "no-taxi") based on the taxi volume, and the disruption period is inferred from the road's status. In other words, we predict the flood period according to the road's status, rather than vice versa. Furthermore, establishing an explicit rule to define "no-taxi" periods may cause confusion, as it implies that we have already observed the field data of flooding and constructed the rule based on it.

- Table 4 - if you had 171 flood gaging sites, why did you only pick 10 to test the model on? Why not test it on all 171?

**Response:**

The data used for parameter calibration were collected in 2015, while the data for method validation were collected in 2019, resulting in a four-year gap between the two datasets due to data availability. Furthermore, Shenzhen, as a coastal city, frequently experienced extreme storm events during summers. To mitigate flooding risks, the Shenzhen Municipal Government annually amends some flood-prone roads. As a result, the hydrological environments of flood-prone roads may have changed during these years, which could render the parameters calibrated based on data from 2015 inaccurate for analysis in 2019. To reduce the validation error caused by this time gap, the roads to be validated should have been vulnerable to flooding in both 2015 and 2019 so that the hydrological parameters of these roads would have a higher chance of remaining unchanged. Approximately 10 roads met this criterion. We have clarified this point in the revised manuscript

(**Line 340**).

- l355 - how did you make a rating curve for each road? How did you get the flow data to relate the stage data to?

**Response:**

The rating curve is usually determined by conducting field measurements and establish the relationship between the observed water level and the corresponding observed flow rate at a measuring point. In this study, however, we had no knowledge of empirical flow data for each road, thus we could not build a real rating curve. Instead, we establish a "rating curve" by plotting the water level which is field measured and the corresponding runoff which is predicted based on the proposed calibration method. If the derived "rating curve" is linearly related, indicating that the predicted runoff has the similar evolution trend of the observed water level, we thus assume that trends of runoff could be correctly predicted. To avoid confusion, we will not use the term "rating curve" to represent the relationship between the predicted runoff and the observed water level in the revised manuscript (**Line 440**).

- Section 4.2 - I personally don't think you need this section. While it's interesting to see how you applied the framework, I don't think it is needed. I think it is enough to have described (section 2), illustrated (section 3), and validated (section 4) the method.

**Response:**

Thank you for your insightful suggestion regarding the section in question. After careful consideration, we agree that the mentioned section may not be necessary for our paper. As you pointed out, the method has been sufficiently described in Section 2, illustrated in Section 3, and validated in Section 4. In response to your suggestion, we will remove the section to streamline the manuscript and maintain focus on the key aspects of our research. We believe this revision will enhance the overall clarity and concision of our paper.

- L54: You might consider citing the following since they are related to this topic (full disclosure: I am an author on both):
   - Sadler, J. M., Goodall, J. L., Morsy, M. M., & Spencer, K. (2018). Modeling urban coastal flood severity from crowd-sourced flood reports using Poisson regression and Random Forest. Journal of hydrology, 559, 43-55.
   - Zahura, F. T., Goodall, J. L., Sadler, J. M., Shen, Y., Morsy, M. M., & Behl, M. (2020). Training machine learning surrogate models from a high-fidelity physics-based model: Application for real-time street-scale flood prediction in an urban coastal community. Water Resources Research, 56, e2019WR027038. https://doi.org/10.1029/2019WR027038

**Response:**

Thank you for your suggestion. We cited this literature in the introduction section to enhance our review of theoretical background (**Line 50**).

Citizens can voluntarily or passively act as human sensors to generate georeferenced data to improve flood monitoring. Many studies have leveraged crowdsourced social media data (Brouwer & Eilander, 2017; Sadler et al., 2018; Zahura et al., 2020), mobile phone data (Yabe et al., 2018; Balistrocchi et al., 2020), and taxi GPS data (She et al., 2019; Kong et al., 2022).

- Figure 10: Could you explain why for some runoff values there is more than one level value? For empirically derived rating curves, each runoff value corresponds to only one water level.

**Response:**

As previously mentioned, the rating curve developed in our study relies on predicted runoff rather than observed runoff. Consequently, the temporal trends of the predicted results may not consistently align with those of the observed water levels. This discrepancy can result in one water level having two distinct runoff values. For instance, consider the road with ID=1 illustrated in Fig.6. When the water level reaches 0.27 m, the corresponding times are 1.1 hour and 1.2 hour. Due to the incongruity between the predicted runoff and observed water level, the runoff values at these two time points are 5.8 m³/s (Point A in Fig. 6) and 12 m³/s (Point B in Fig. 6), which accounts for the presence of two runoff values for a single water level.

[Figure]

**Figure 6** An example to show why some runoff values correspond to two level values.

**Editorial comments**

**Response:**

Modified as suggested (**Line 23**).

**Response:**

Modified to "incomplete" as suggested (**Line 31**).

**Response:**

Modified as suggested (**Line 59**).

**Response:**

The definition of hydrograph is removed (**Line 88**).

**Response:**

Modified to "a certain amount of water" (**Line 146**).

- l154 - I suggest changing "converts rainfall excess to direct runoff" to "converts rainfall excess to a temporal distribution of direct runoff" or something like that to communicate that it is a distribution of runoff over time.

**Response:**

Modified as suggested (**Line 159**).

- l160 - "the paucity of runoff" instead of "the paucity of the runoff"

**Response:**

Modified as suggested (**Line 159**).

- l161 - "sparkled" is probably not the right word here. Maybe "sparked" or "motivated"

**Response:**

Modified to "motivated" as suggested (**Line 167**).

**Response:**

Modified as suggested (**Line 207**).

**Response:**

Modified as suggested (**Line 212**).

**Response:**

Modified as suggested (**Line 214**).

- Table 1: Is it correct to have the "/"s for Feature in several of the rows? If so, maybe you should define that means.

**Response:**

Modified the "/" to "Not mention" in Table 1 (**Table 1**).

- l295 - suggest changing "a little bit" to "slightly" or something similar. "a little bit" is imprecise and colloquial

**Response:**

Modified to "slightly" as suggested (**Line 317**).

- l308 - "waterlogging" is not a term I typically hear. Do you mean something like "flood-prone?"

**Response:**

Modified to "flood-prone" to enhance clarity (**Line 263**).

- Figure 9 - is the x-axis "Time of Concentration?" If so, please change. I didn't know what "Time" meant.

**Response:**

Modified to "Time of Concentration" to enhance clarity (**Figure 9**).

- l396 - suggest replace "great" with "good"

**Response:**

Modified as suggested (**Line 474**).

- l434 - suggest remove "great" to read "This study illustrates the potential ... "

**Response:**

Modified as suggested (**Line 513**).

**Response to Reviewer 2**

**Summary**

The paper presents a novel approach for calibrating an urban rainfall-runoff model using taxi GPS data. This is an original idea that seems to have potential as demonstrated in this study. I also commend the authors for making available their data and code.

**Response:**

We thank the reviewer for the constructive comments to help us improve the manuscript. We are pleased that the manuscript aroused the reviewer's interest and are thankful for the positive feedback. Our responses to your comments are provided below. The data and code used to validate the method are available at Zenodo (https://doi.org/10.5281/zenodo.7894921).

**Major comments**

-What's the reason for modeling the taxi data as pass/no pass instead of directly modeling the number of taxis passing? The latter somehow seems more obvious since the original data are taxi counts, while your approach first requires converting taxi counts to 0/1 values, which introduces a potential loss of information. Please better justify this modeling choice.

**Response:**

We attempted to utilize taxi counts as an indicator of road status. However, establishing a relationship between precipitation or runoff and taxi count proved to be challenging. In situations where the road is not entirely disrupted by runoff, a stable quantitative relationship between taxi count and precipitation or runoff is hard to capture, as most taxis do not alter their travel routes when the water level is not too high. Consequently, estimating $\mathcal{L}(X|\Omega^{(i)})$ becomes difficult when $X$ represents taxi volume. In contrast, when the road is disrupted, the taxi volume should be zero by the definition of disruption, and typically greater than zero when the road is open, simplifying the estimation of $\mathcal{L}(X|\Omega^{(i)})$. Overall, if the method calibrates parameters based on the taxi count, it may select the optimal parameter that best corresponds to the observed taxi count rather than the road disruption status, thereby introducing unnecessary errors.

-An alternative approach would be to use the Poisson distribution to directly model the number of taxis passing (rather than arriving). Have you considered this? This would be more like a Poisson regression model, but perhaps leveraging your road disruption function to model lambda instead of the usual Poisson link function.

**Response:**

We appreciate your suggestion. As stated in our previous response, this approach requires establishing a numerical relationship between the road disruption function and lambda, which denotes the mean of the 5 min taxi volume. Since most taxis do not modify their travel routes during less intense runoff, the effect of rainfall on taxi volume becomes difficult to capture.

-Did you check whether the Poisson model for the number of taxis arriving at a road is a good assumption for your data?

**Response:**

In the revised manuscript (**Line 496**), a Chi-square goodness of fit test is conducted to check whether the frequency distribution adheres to a Poisson distribution. The squared differences between the observed and expected 5 min taxi frequencies, as predicted by the Poisson distribution, are computed to construct the Chi-square statistic. This statistic is then compared to the critical value corresponding to a significance level of 0.05. If the Chi-square statistic exceeds the critical value, the null hypothesis—that the probability distribution follows a Poisson distribution—is rejected; otherwise, the null hypothesis is accepted, suggesting the distribution may conform to a Poisson distribution.

Ten roads illustrated in Figure 8 of the manuscript were selected, and the frequency distribution of 5 min taxi volume for each period on each road was derived. Each frequency distribution consists of 31 samples, representing the taxi volume during a specific 5 min period collected from May 1, 2015, to May 31, 2015, for a specific road. The Chi-square goodness of fit test was applied to each frequency distribution, and the proportion of periods following a Poisson distribution for each road was calculated. The test results are presented in Table 1. According to these results, the frequency distribution of the 5 min taxi volume during a specific period adheres to the Poisson distribution for more than 50% of the periods in 7 of the 10 roads. Consequently, the Poisson model appears to be a suitable assumption in this study.

**Table 1** Chi-square goodness of fit test of Poisson distribution for 10 roads.

| Road ID | 1 | 2 | 3 | 4 | 5 | 6 | 7 | 8 | 9 | 10 |
|---|---|---|---|---|---|---|---|---|---|---|
| Proportion of periods that follow Poisson distribution % | 12.5 | 95.8 | 79.2 | 97.5 | 85.8 | 27.5 | 96.7 | 48.3 | 73.3 | 95.8 |

-A limitation is that all variables are treated as discrete random variables whereas the hydrological model parameters are continuous. Why discretize the parameters?

**Response:**

The reason to discretize parameters stems from the challenges associated with solving optimal problems. While continuous parameters may yield more accurate estimations, it is often arduous to obtain an analytical or numerical solution for $\boldsymbol{\theta}^{(i)}$ that maximizes $P(\boldsymbol{\theta}^{(i)})\mathcal{L}(\boldsymbol{X}|\boldsymbol{\theta}^{(i)})$ from a continuous parameter space. For instance, finding the analytical solution which maximizes $P(\boldsymbol{\theta}^{(i)})\mathcal{L}(\boldsymbol{X}|\boldsymbol{\theta}^{(i)})$ necessitates differentiating $\mathcal{L}(\boldsymbol{X}|\boldsymbol{\theta}^{(i)})$, which may not always be feasible.

Given a catchment area and a time of concentration, constructing a synthetic unit hydrograph (SUH) based on the SCS unit hydrograph is straightforward. However, determining these two parameters from a given SUH poses a significant challenge. Similarly, generating runoff by combining rainfall with the SUH through a convolution formula is easy, but deriving the SUH from rainfall and runoff is difficult. Chow et al. (1988) demonstrated deconvolution methods such as matrix calculations or linear programming to derive the SUH, but these approaches are complex and cannot provide explicit functions that input runoff and output SUH. To circumvent the backward parameter-solving process, we discretize continuous parameters and calculate $P(\boldsymbol{\theta}^{(i)})\mathcal{L}(\boldsymbol{X}|\boldsymbol{\theta}^{(i)})$ for each parameter set using a forward calculation process, which is more convenient in this study. It is important to note that the exclusion of continuous parameters in this study is due to the complexity of differentiating the proposed three-step procedure, not an indication that they are inapplicable to other procedures.

-Does the model account for other (non-flooding) factors that may affect the number of taxis

in a road, e.g. time of day (rush hour)?

**Response:**

The time-of-day variation may affect the taxi volume in a road. In the previous version of our manuscript, we did not account for variations in taxi volume concerning the time-of-day or day-of-week. We assumed that the average number of taxis arriving on the road was constant, and the no-taxi-passing probability is given by:

$$\omega_t^{(i)} = e^{-\lambda_t} \sum_{n=0}^{\infty} \left( P(Disrupt)_t^{(i)} \lambda \right)^n / n! = \exp\left( \lambda \left( P(Disrupt)_t^{(i)} - 1 \right) \right) \qquad (1)$$

where $\lambda$ is the average taxi volume per 5 min interval, calculated by averaging all 5 min taxi volumes using historical taxi GPS data for a specific road.

However, the value of $\lambda$ fluctuates according to the time of day, indicating higher taxi volume during congested periods and lower volume during non-congested periods. In the revised manuscript (**Line 245**), we incorporated the time-of-day variation in taxi volume when computing the no-taxi-passing probability:

$$\omega_t^{(i)} = e^{-\lambda_t} \sum_{n=0}^{\infty} \left( P(Disrupt)_t^{(i)} \lambda_t \right)^n / n! = \exp\left( \lambda_t \left( P(Disrupt)_t^{(i)} - 1 \right) \right) \qquad (2)$$

where $\lambda_t$ is the average number of taxis arriving at the road during the $t$th interval. Compared with $\lambda$, $\lambda_t$ has smaller deviance because it excludes more non-flooding factors.

-Curve number CN is kept fixed even though it is also uncertain.

**Response:**

Thank you for pointing this out. We acknowledge that fixing the curve number as 85 is not realistic as it is influenced by various factors in urban areas, such as impervious surface percentage and soil type. Therefore, we have revised the manuscript to include curve number as one of the parameters to be calibrated (**Line 375**). In total, we calibrate three parameters: catchment area, time of concentration, and curve number.

[revised manuscript text omitted]

**Response:**

Thanks for your suggestion. The Bayes equation is rewritten as (**Line 105**):

$$P(\boldsymbol{\theta}^{(i)}|X) = P(\boldsymbol{\Omega}^{(i)}|X) \propto P(\boldsymbol{\theta}^{(i)})\mathcal{L}(X|\boldsymbol{\theta}^{(i)}) \tag{3}$$

**Editorial comments**

-eq. 11: please define x and y

**Response:**

The expression of the fitting curve is:

$$y = [1 + \exp(-16.6(x - 0.48)^2)]^{-1} \tag{4}$$

where $x$ is the product of flow velocity and flow depth, and $y$ is the disruption probability (**Line 223**).

-L23: metropolis --> metropolises or metropolitan areas

**Response:**

Modified as suggested (**Line 23**).

-L40: "calibrated on runoff data alone" - there are many studies that calibrate on other data as well
Response:

**Response:**

It is not rigorous to say that "No matter what kinds of methods, hydrological models are basically calibrated based on the runoff data alone." Thus, that sentence is removed (**Line 40**).

-L47: ungaged vs ungauged: pick one spelling

**Response:**

Modified to "ungauged" as suggested.

-L83 (and other places): equals to --> equals

**Response:**

Modified as suggested.

-L90: arriving --> arrival

**Response:**

Modified as suggested.

-L99: does index i refer to road i?

**Response:**

No, the index $i$ refers to the $i$th parameter set (**Line 99**).

-L132: instantization --> instantiation

**Response:**

Modified as suggested (**Line 136**).

-suggest to proofread entire manuscript to fix issues with use of English

**Response:**

We thank the reviewer for pointing this out. As suggested, the manuscript is thoroughly proofread, and the grammar, clarity, and overall readability is also improved.

---

## Author Response (AR2)

**Report 1**

We express our gratitude to the reviewer for the suggestion, which is rigorous and improves the quality of our manuscript. Following careful consideration, we have amended the manuscript in accordance with your valuable suggestion. Our response to your suggestion is provided below.

- I'm satisfied with the responses provided by the authors. Perhaps one minor point is to explicitly acknowledge the limitation in the manuscript of discretizing the parameter values, noting that this is not strictly necessary since there are established methods for solving the optimization/posterior inference problem in continuous space (e.g. Monte Carlo) without the need for differentiation (as the authors argued in their response).

**Response:**

In the revised manuscript, we acknowledged the limitation of discretizing the parameter values, and offered additional recommendations accordingly **(Line 503)**:

Fourth, it is imperative to acknowledge that the parameter values in this study were discretized, although hydrological model parameters are inherently continuous. This discretization approach could result in the omission of optimal solutions, particularly when hydrological models exhibit sensitivity to these parameters. It is important to note that discretization is neither a requisite nor a recommended strategy. Future research should address the optimization or posterior inference problem in a continuous parameter space based on established methods such as the Monte Carlo algorithm.

**Report 3**

We express our gratitude to the reviewer for the editorial suggestions, which will very improve the quality of our manuscript. Following careful consideration, we have amended the manuscript according to your valuable comments. Our responses to your comments are provided below.

**Editorial comments:**

- I suggest changing the first sentence of the abstract to, "Hydrological parameters should pass through a careful calibration procedure before being used in a hydrological model that aids decision making." That way it is clear the hydrological parameters are used in the models which are in turn used for decision making. The "parameters" aren't used for decision making typically in my understanding.

**Response:**

Modified as suggested (**Line 12**).

- line 16 - suggest "corresponded" (instead of "corresponds) to keep the tense consistent through the sentence"

**Response:**

Modified to "corresponded" as suggested (**Line 17**).

- line 29 - suggest "Hydrologic modeling *has* a relatively well-established theory" or maybe *is based on* or something similar. Just to communicate more clearly since modeling is not theory but is based on theory

**Response:**

Added "based on" as suggested (**Line 30**).

- line 42 - suggest changing "they have no motivation" to "these cities are not always motivated"

**Response:**

Modified as suggested (**Line 43**).

- line 73 - suggest changing "we have several hydrological models" to "we have several versions of a hydrological model" so that people know you it's not like you have SWAT, HEC-HMS, CN, etc.

**Response:**

Modified as suggested (**Line 74**).

- Several of the figures were quite pixilated. I assume that will be worked out by the copy editors, but bringing it up just in case.

**Response:**

We will improve the figure quality in the next process according to the suggestion of copy editors.